Report

EMBO
reports

# S-acylation of a non-secreted peptide controls plant immunity via secreted-peptide signal activation

Wenliang Li [1,3], Tushu Ye [1,3], Weixian Ye [1,3], Jieyi Liang [1], Wen Wang [2], Danlu Han [1], Xiaoshi Liu [1], Liting Huang [1], Youwei Ouyang [1], Jianwei Liao [1], Tongsheng Chen [2], Chengwei Yang [1✉] & Jianbin Lai [1✉]

## Abstract

Small peptides modulate multiple processes in plant cells, but their regulation by post-translational modification remains unclear. ROT4 (ROTUNDIFOLIA4) belongs to a family of Arabidopsis non-secreted small peptides, but knowledge on its molecular function and how it is regulated is limited. Here, we find that ROT4 is S-acylated in plant cells. S-acylation is an important form of protein lipidation, yet so far it has not been reported to regulate small peptides in plants. We show that this modification is essential for the plasma membrane association of ROT4. Overexpression of S-acylated ROT4 results in a dramatic increase in immune gene expression. S-acylation of ROT4 enhances its interaction with BSK5 (BRASSINOSTEROID-SIGNALING KINASE 5) to block the association between BSK5 and PEPR1 (PEP RECEPTOR1), a receptor kinase for secreted plant elicitor peptides (PEPs), thereby activating immune signaling. Phenotype analysis indicates that S-acylation is necessary for ROT4 functions in pathogen resistance, PEP response, and the regulation of development. Collectively, our work reveals an important role for S-acylation in the cross-talk of non-secreted and secreted peptide signaling in plant immunity.

Keywords Arabidopsis; Plant Immunity; Protein S-acylation; ROT4; Small Peptide
Subject Categories Immunology; Plant Biology; Signal Transduction

## Introduction

Small peptides, which are proteins with less than 100 amino acid residues, are emerging as critical regulators in plant development and stress responses (Takahashi et al, 2019). Small peptides are classified into two groups, secreted peptides and non-secreted peptides. Secreted peptides work as a ligand for binding the extracellular domain of receptors for downstream signaling transduction (Murphy et al, 2012). However, the molecular functions of non-secreted peptides in plant cells remain unclear.

The ROT-FOUR LIKE/DEVIL (RTFL/DVL) family is a group of non-secreted small peptides widely conserved in land plants (Guo et al, 2015). Previous studies have shown that the overexpression of members of this family in Arabidopsis, such as *ROTUNDIFOLIA4* (*ROT4*) or *DEVIL1* (*DVL1*)/*RTFL18*, results in a phenotype with small and round leaves (Narita et al, 2004; Wen et al, 2004). The overexpression of *ROT4* reduces the number of leaf cells along the longitudinal axis and has severe effects on the development of trichomes and siliques (Ikeuchi et al, 2011; Valdivia et al, 2012). In Arabidopsis, 23 potential RTFL/DVL members were identified with a conserved domain of around 30 residues, possibly playing redundant roles in development regulation (Narita et al, 2004). The interaction partner of RTFL/DVL members remains to be identified, and thus the molecular mechanism of how this small peptide family works in plant cells is unclear.

In the current study, we determined the function of ROT4 in the activation of plant immunity. Small peptides are involved in the regulation of plant immunity, yet the majority of previous research focuses on secreted peptides in this process (Segonzac and Monaghan, 2019). For example, PAMP-induced peptides (PIPs) are a group of secreted peptides that bind to RECEPTOR-LIKE KINASE 7 for immune modulation (Hou et al, 2014). Although Plant elicitor peptides (PEPs) have been expressed without an N terminus signaling peptide motif which mediates conventional secretion, the mature forms of these peptides are also secreted outside and recognized by PEP RECEPTOR1/2 (Bartels and Boller, 2015). Thus, these secreted peptides work as ligands of their receptors in the extracellular space for immune regulation, yet the mechanism of how non-secreted peptides work in plant immunity remains to be studied.

Although previous work has shown that ROT4 localizes on the plasma membrane, no transmembrane domain is predicted in this peptide (Narita et al, 2004). Thus, it is important to investigate how ROT4 associates with the plasma membrane and to determine whether this localization is essential for its function. Here, we found that ROT4 is S-acylated for its membrane localization in plant cells. S-acylation, also denoted S-palmitoylation, is an important post-translational modification that transfers long-chain fatty acids onto the cysteine residues of substrates to regulate their localization and functions (Linder and Deschenes, 2007). For instance, the formation of receptor kinase complex in plant immunity is

[1]Guangdong Provincial Key Laboratory of Biotechnology for Plant Development, School of Life Science, South China Normal University, Guangzhou 510631, China. [2]Key Laboratory of Laser Life Science, MOE Key Laboratory of Laser Life Science, College of Biophotonics, South China Normal University, Guangzhou 510631, China. [3]These authors contributed equally: Wenliang Li, Tushu Ye, Weixian Ye. ✉E-mail: yangchw@scnu.edu.cn; 20141062@m.scnu.edu.cn

precisely modulated by S-acylation (Hurst et al, 2023). Several post-translational modifications are essential for the function of small peptides, such as tyrosine sulfation, proline hydroxylation, and hydroxyproline arabinosylation (Stuhrwohldt and Schaller, 2019). However, the function of S-acylation on small peptides has not yet been characterized in plant cells.

In this study, we find that the non-secreted small peptide ROT4 is S-acylated to promote its membrane association in plant cells. This modification of ROT4 mediates its interaction with the cytoplasmic kinase BSK5 to interfere with the binding of BSK5 to the receptor kinase PEPR1, which perceives secreted PEPs for the activation of plant immunity (Huffaker and Ryan, 2007). Therefore, this study establishes a connection between non-secreted and secreted peptide signaling mediated by protein S-acylation for plant immune regulation. The results expand the field of post-translational modifications of plant small peptides and might aid in the genetic improvement of pathogen resistance in crops.

## Results and discussion

### S-acylation mediates the plasma membrane localization of ROT4

Although ROT4 has been shown to be localized on the plasma membrane (Narita et al, 2004), how and why ROT4 is associated with the plasma membrane has remained unanswered for around two decades. Peptides can undergo several types of post-translational modifications (Matsubayashi, 2014), and thus we used bioinformatics methods to analyze potential modifications on ROT4 for the regulation of its localization and function. Using CSS-PALM analysis (Ren et al, 2008), several cysteine residues on the ROT4 protein were predicted as potential sites of S-acylation (Fig. EV1A), which typically mediates the membrane association of target proteins. Therefore, the S-acylation of the wild-type and site-mutated (from cysteine to serine) variants of GFP-fused ROT4 were detected in a biotin-switch assay. Different versions of GFP-ROT4 were expressed in protoplasts and free cysteine residues of proteins in lysates were chemically blocked; S-acyl groups were specifically removed via hydroxylamine treatment to release the previously S-acylated cysteine residues; after cysteine conjugation with biotin, the targeted proteins were captured on streptavidin resin and eluted by reducing reagents. The results indicate that the wild-type GFP-ROT4 was S-acylated in plant cells, and the majority of these cysteine mutations did not affect its S-acylation. However, the mutation of C42 dramatically reduced the S-acylation level (Fig. 1A,B), suggesting that C42 is the predominant S-acylation site on ROT4.

To determine the contribution of S-acylation on the plasma membrane association of ROT4, the subcellular localization of the wild-type and mutant versions of GFP-ROT4 were analyzed in protoplasts. The mutation of C9, C12, C19, or C36 did not alter the plasma membrane localization of GFP-ROT4, yet the C42 mutation resulted in the diffusion of a proportion of GFP-ROT4 into the cytoplasm (Fig. 1C). The cell fractionation data indicated that the majority of the wild-type GFP-ROT4 was accumulated in the membrane fraction and a high proportion of the C42S version of GFP-ROT4 was found in the soluble fraction (Fig. 1D; controls are shown in Fig. EV2A). The observation of GFP-ROT4 in transgenic

plants also confirmed that the C42S mutation altered its localization in guard cells and root cells (Fig. 1E,F). These findings indicate that S-acylation is essential for the plasma membrane localization of ROT4 in plant cells.

The Arabidopsis RTFL family contains 23 members and we attempted to determine the conservation of the C42 site of ROT4 among the RTFL proteins. The protein alignment results indicate that this cysteine residue can be found in most RTFL members, except RTFL11, RTFL13, and RTFL14 (Fig. EV1B). To provide evidence for the conserved function of S-acylation on RTFL proteins, we selected RTFL13 (without this cysteine residue) and RTFL18 (with this cysteine residue) for further analysis (Fig. 1G). GFP-RTFL13 was observed to be distributed globally in the cells and GFP-RTFL18 was localized on the plasma membrane (Fig. 1H). The biochemical results reveal that GFP-RTFL18 was S-acylated in plant cells, but this was not the case for GFP-RTFL13 (Fig. 1I). The mutation of the arginine residue at this position to cysteine residue enhanced the membrane association of RTFL13 (Fig. EV2B,C). Consistently, the overexpression of RTFL18 but not RTFL13 resulted in an abnormal developmental phenotype (Fig. EV2D–F). These results indicate that this conserved S-acylated site is critical for the plasma membrane association of RTFL members. Given that the S-acylated cysteine residue of ROT4 can be found in most RTFL members, S-acylation may be a general regulation strategy for the members in this family and even their homologs in other species.

### Induction of immune-responsive genes via ROT4 requires its S-acylation

Given that the RTFL family contains 23 members with redundant functions, overexpression is generally employed to investigate the physiological roles of RTFL members (Guo et al, 2015). Previous studies showed that the overexpression of RTFL genes, such as ROT4, alters the developmental phenotypes of Arabidopsis (Narita et al, 2004; Wen et al, 2004), yet the molecular mechanism of how these peptide members work in plant cells remains to be studied. To investigate the pathways regulated by ROT4, RNA was extracted from the wild-type control and wild-type ROT4 overexpressing plants (the verification of ROT4 expression in the ROT4(WT) overexpressing plants is included in Appendix Fig. S1) for RNA-Seq analysis. The RNA-Seq data indicated that over 800 genes were upregulated and less than 100 genes were downregulated in the wild-type ROT4 overexpressing plants, compared to the wild-type control plants (Fig. 2A; Appendix Fig. S2A; Dataset EV1). Surprisingly, the Gene Ontology (GO) analysis indicated that the differentially expressed genes (DEGs) were enriched in the "Response to salicylic acid" and other pathogen resistance pathways (Fig. 2B; Dataset EV2). The KEGG analysis also indicated that the DEGs were enriched in the "plant-pathogen interaction" pathway (Appendix Fig. S2B). The heat map of the DEGs in "plant-pathogen interaction" showed the upregulation of genes in the ROT4 overexpressing plants (Fig. 2C). The upregulation of immune genes in the wild-type ROT4 overexpressing lines may explain their developmental defects via constitutive immune activation, which typically interferes with normal growth.

To determine the effect of S-acylation on the function of ROT4, the expressions of immunity-related genes were measured in two independent wild-type ROT4 overexpressing lines and two independent C42S version of ROT4 overexpressing lines with

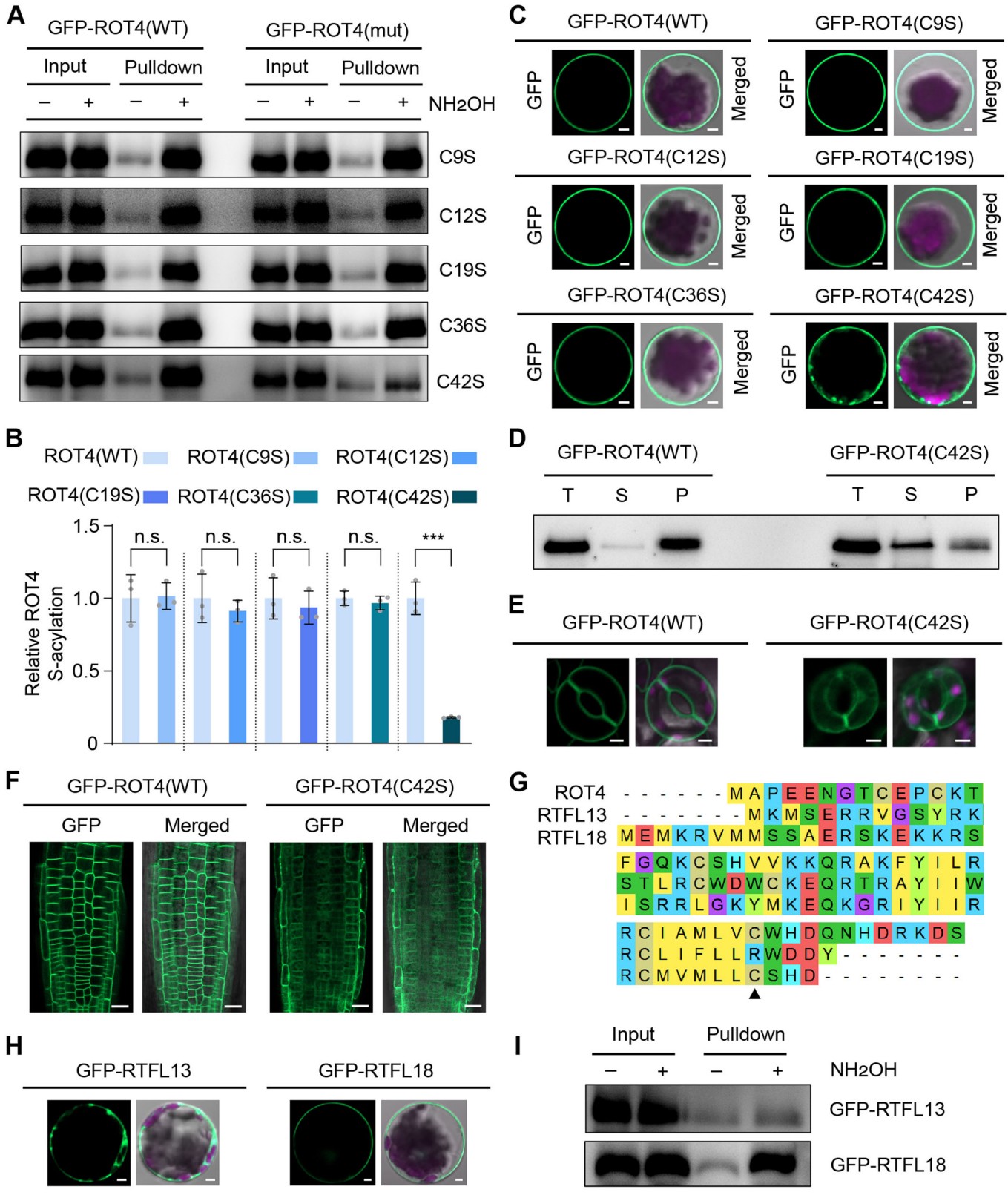

**Figure 1.  S-acylation mediates the plasma membrane localization of ROT4.**

(A, B) Identification of the S-acylation sites of ROT4 in plant cells. The wild-type (WT), C9S, C12S, C19S, C36S, and C42S versions of GFP-ROT4 were expressed in protoplasts. Their S-acylation levels were measured in biotin-switch assays. Cell lysates are shown in the input samples and the S-acylated proteins are shown in the pulldown samples dependent on $NH_2OH$. The anti-GFP immunoblots are representative of three biologically independent experiments (A). The immunoblot signals were quantified by ImageJ and modification levels were calculated from relative signals ([pulldown+/input+] − [pulldown−/input−]). The relative S-acylation level of ROT4(WT) was set to 1. The statistical data from three biologically independent experiments are shown in (B). (C) The function of S-acylation for subcellular localization of ROT4 in protoplasts. The wild-type and mutant versions of GFP-ROT4 were expressed in protoplasts, and 24 h after transformation, the GFP-ROT4 localization was detected using confocal microscopy. The representative GFP (green) and merged (with the bright field in gray and chloroplast auto-fluorescence in magenta) signals from three biologically independent experiments are shown. Scale bars: 5 μm. (D) Comparison of membrane association of the wild-type and C42S versions of ROT4 in a cell fractionation assay. Total proteins (T) from protoplasts were divided into pellet (P) and soluble (S) fractions via ultra-centrifugation. The representative anti-GFP immunoblot from three biologically independent experiments is shown. The cytosol and membrane fraction controls are included in Fig. EV2A. (E, F) Effect of the C42S mutation on ROT4 localization in guard cells and root cells. Six-day-old transgenic seedlings expressing the WT or C42S version of GFP-ROT4 were used for analysis via confocal microscopy. The representative GFP (green) and merged (with the bright field in gray and chloroplast auto-fluorescence in magenta) signals in guard cells (E) and root cells (F) from three biologically independent experiments are shown. Scale bars: 5 μm in (E) and 20 μm in (F). (G) Protein sequence alignment of ROT4, RTFL13, and RTFL18. The sequences were analyzed by ClustalW and the C42 residue in ROT4 is indicated. (H) The subcellular localization of GFP-RTFL13 and GFP-RTFL18. The representative GFP (green) and merged (with the bright field in gray and chloroplast auto-fluorescence in magenta) signals in the protoplasts from three biologically independent experiments are shown. Scale bars: 5 μm. (I) Detection of S-acylation levels of RTFL13 and RTFL18. The proteins expressed in protoplasts were used in biotin-switch assays. The representative anti-GFP immunoblots from three biologically independent experiments are shown. Data information: In (B), data are presented as mean ± standard deviation (SD). ***$P < 0.001$, n.s., not significant ($P > 0.05$), Student's *t*-test (two-tailed). Source data are available online for this figure.

similar expression levels of *ROT4* (Appendix Fig. S1), compared to those in the wild-type and *GFP* vector controls. Based on the RNA-Seq data, the transcript levels of eight genes related to plant immunity were detected using quantitative RT-PCR. Compared with those in the control plants, the expression of these genes was dramatically upregulated in the wild-type *ROT4* overexpressing lines, but not in the C42S version of *ROT4* overexpressing lines (Fig. 2D). This result suggests that the S-acylation-mediated localization is critical for the function of ROT4 in immune activation.

## S-acylation of ROT4 enhances its association with BSK5 for immune activation

To uncover the molecular function of ROT4, immunoprecipitation (IP) and mass spectral analysis were performed on the *GFP* vector control and *GFP-ROT4* plants to identify ROT4-interacting proteins (Appendix Fig. S3A). The mass spectral analysis identified 713 proteins that were specifically associated with GFP-ROT4 (Appendix Fig. S3B; Dataset EV3). Given that the plasma membrane localization mediated by S-acylation is critical for the function of ROT4, its interacting partners on the plasma membrane from a GO analysis were potential targets for downstream signaling transduction (Appendix Fig. S3C; Dataset EV4). In these ROT4-interacting proteins on the plasma membrane, BSK5, a receptor-like cytoplasmic kinase involved in plant immunity (Majhi et al, 2019), was selected as a candidate for further analysis. The expression of *PR1* was dramatically upregulated in both the *BSK5* mutant (Majhi et al, 2019) and *ROT4* overexpressing plants, suggesting their potential association in plant immunity.

As S-acylation is necessary for ROT4 localization on the plasma membrane, we evaluated the effect of C42S mutation on its interaction with BSK5. The co-immunoprecipitation (co-IP) data of ROT4 and BSK5 with different tags indicated that the wild-type ROT4 interacted with BSK5, while the C42S mutation reduced their association (Fig. 3A–C). The fluorescence resonance energy transfer (FRET) and bimolecular fluorescence complementation (BiFC) results confirmed that the C42S mutation disrupted the interaction between ROT4 and BSK5 in plant cells (Figs. 3D and EV3). These

results suggest that the S-acylation of ROT4 is important for its association with BSK5. Our recent work indicated the BSK members are also S-acylated (Liu et al, 2023), and thus S-acylation on both ROT4 and BSK5 may contribute to their association in the same membrane regions in plant cells.

Previous research has shown that BSK5 associates with the receptor kinase PEPR1 on the plasma membrane in the regulation of plant immunity (Majhi et al, 2019). Hence, our data on the ROT4-BSK5 interaction implies that ROT4 may be associated with the PEP signaling pathway. Therefore, we tested the effect of Pep1 on the interaction between ROT4 and BSK5. The Pep1 treatment was observed to increase the ROT4-BSK5 association (Fig. 3E), supporting the functional interplay between ROT4 and the PEPR1-mediated signaling. Thus, the effect of ROT4 on the BSK5-PEPR1 interaction was further measured in a co-IP assay. Compared to that in the wild-type control plant cells, the interaction between BSK5-Myc and PEPR1-GFP was attenuated in the wild-type *ROT4* (without a tag fusion) overexpressing transgenic plant cells (Fig. 3F), suggesting that ROT4 enhances the dissociation between BSK5 and PEPR1.

After the perception of extracellular signals, the receptor kinases may phosphorylate their associated cytoplasmic kinases, resulting in the release of cytoplasmic kinases for downstream immune activation. For example, flg22 induces the dissociation between its receptor kinase FLS2 and BSK1 (Shi et al, 2013), the homolog of BSK5, for downstream immune responses. Thus, to determine whether Pep1 affects the PEPR1-BSK5 association, Pep1 was applied to the co-IP assay. The results indicate that the secreted peptide Pep1 also reduced the interaction between PEPR1-GFP and BSK5-Myc (Fig. 3G). The quantification of interaction intensity in co-IP confirmed that the ROT4-BSK5 interaction was enhanced by Pep1; however, the PEPR1-BSK5 interaction was decreased by Pep1, which was similar to the effect of ROT4 overexpression (Fig. 3H). Previous research has shown that the phosphorylation of BSK5 mediated by PEPR1 is critical for the function of BSK5 in immune responses (Majhi et al, 2019). Our co-IP data showed that the overexpression of *ROT4* or Pep1 treatment attenuated the interaction between PEPR1 and BSK5(WT) but did not affect the association between PEPR1 and BSK5(S209A/T210A)

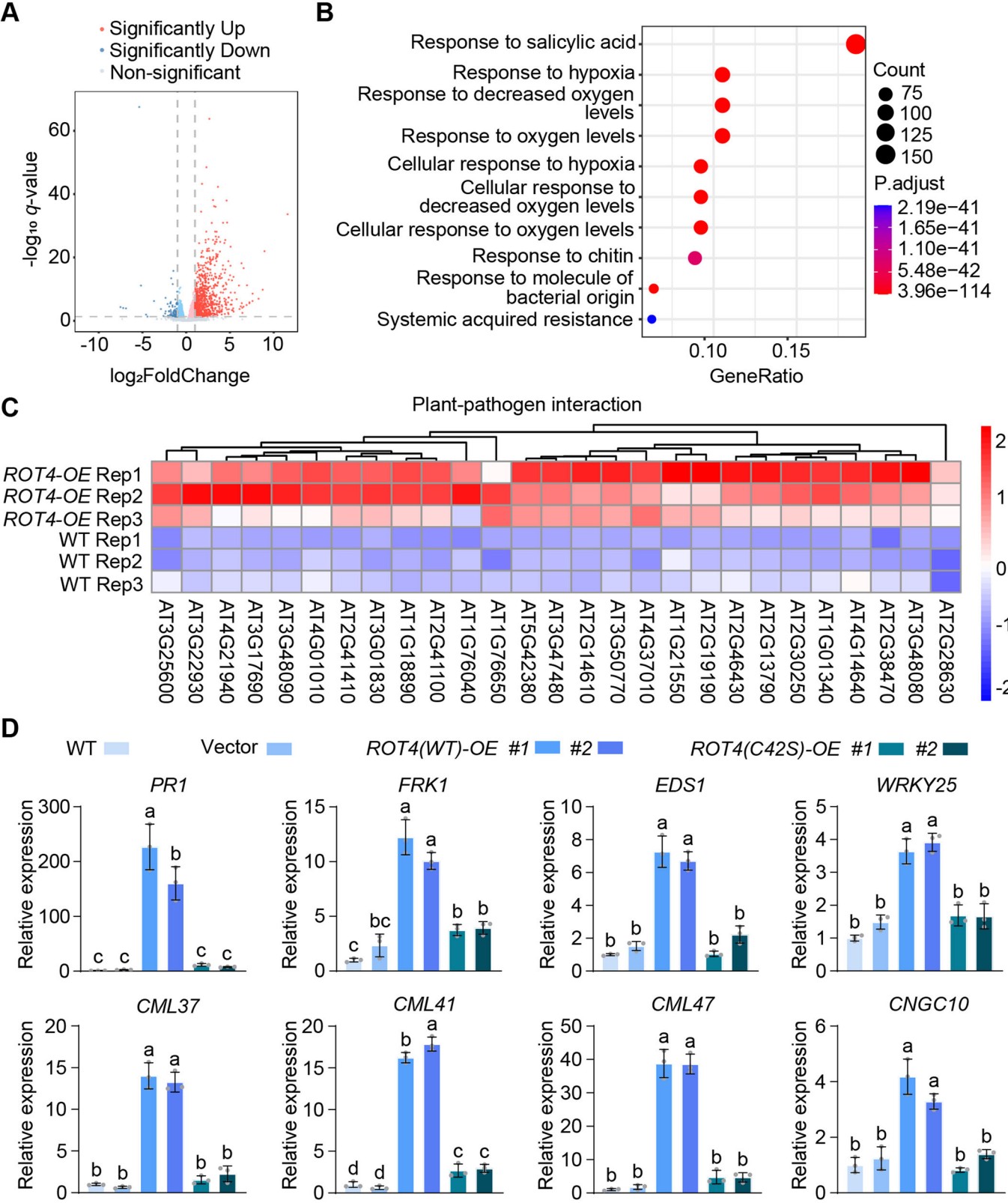

**Figure 2. Induction of immune-responsive genes via ROT4 requires its S-acylation.**

(A) Effect of *ROT4* overexpression on the gene expression in Arabidopsis. The volcano plot for the differentially expressed genes (DEGs) in the wild-type control and *GFP-ROT4* overexpressing plants is shown. FDR < 0.01, |log$_2$fold change| > 1. (B) Processes in which gene expression was regulated by *ROT4* overexpression. The biological process enrichment of DEGs in GO analysis is shown. *P* < 0.05. (C) Expression alteration of genes related to plant-pathogen interaction mediated by overexpression of *ROT4*. The heat map of the relative expression of DEGs in plant-pathogen interaction from three biologically independent replicates is shown. (D) Effect of S-acylation of ROT4 on its function in the regulation of immune gene expression. Transcript levels of the wild-type, *GFP* vector control, two independent wild-type *GFP-ROT4* overexpressing lines, and two independent C42S versions of *GFP-ROT4* overexpressing lines were measured via quantitative RT-PCR. *ACTIN2* was used as an internal control. The relative expression levels in the WT plants were set to 1. Data information: In (D), data are presented as mean ± SD from triplicated technological repeats in an experiment; the expression patterns were consistent in three biologically independent experiments. Significance was analyzed via one-way ANOVA followed by Tukey's multiple comparison tests (*P* < 0.05). Source data are available online for this figure.

(Fig. EV4A–D), which is a phosphorylation defective mutant. This suggests that the phosphorylation of BSK5 is necessary for its dissociation from PEPR1 induced by Pep1 and ROT4. Thus, the dissociation of PEPR1 and BSK5 mediated by S-acylated ROT4 mimics the process in the presence of Pep1, activating downstream immune pathways.

## S-acylation is essential for the function of ROT4 in plant immunity and pathogen resistance

Given that our data suggested that overexpression of *ROT4* enhances plant immunity, the dwarf phenotype of the *ROT4* overexpressing plants may be a result of constitutively activated immune signaling. Therefore, the phenotypes of the wild-type and C42S versions of *ROT4* overexpressing plants were analyzed. The result showed that overexpression of the wild-type but not the C42S version of *GFP-ROT4* suppressed normal development, exhibiting a dwarf phenotype with small and round leaves (Fig. 4A). Consistently, overexpression of the wild-type *ROT4* did not result in development suppression in the *PEPR1* mutants (Fig. EV5), supporting the notion that ROT4 is associated with PEP immune signaling.

As the gene expression analysis indicated that immunity was activated in the wild-type *ROT4* overexpressing lines, pathogen infections were used to evaluate the function of S-acylated ROT4 in disease resistance. The inoculation of a bacterial pathogen *Pseudomonas syringae* pv. *tomato* DC3000 showed that the accumulation of leaf bacteria was significantly lower in the wild-type *ROT4* overexpressing lines compared to the C42S version of *ROT4* overexpressing lines and control plants (Fig. 4B). Similarly, the overexpression of the wild-type version but not the C42S version of *ROT4* attenuated symptoms of the infection from the fungal pathogen *Botrytis cinerea*, either with a pathogenic peptide flg22 (for exogenous induction of immune responses) or an endogenous peptide Pep1 (Fig. 4C,D). These data indicate that the S-acylated ROT4 increases the resistance of plants against pathogens.

The S-acylation of ROT4 is critical for its interaction with BSK5 for the dissociation of the PEPR1-BSK5. Based on this, we evaluated the effect of exogenous Pep1 on plant development. Under the Pep1 treatment, the suppression of root development in the wild-type *ROT4* overexpressing lines was significantly larger than that of the control plants and the C42S version of *ROT4* overexpressing lines (Fig. 4E,F). This suggests that the Pep1-PEPR1 immune pathway can be activated more easily during the over-accumulation of S-acylated ROT4.

In general, BSK5 and PEPR1 form a dimer under a resting state in normal conditions, while Pep1 binding to PEPR1 releases BSK5 from the complex, and excess S-acylated ROT4 interacts with BSK5 to dissociate the PEPR1-BSK5 complex for constitutive activation of immunity (Fig. 4G). As BSK5 belongs to the kinase family involved in BR signaling transduction (Sreeramulu et al, 2013), the interplay between BSK5 and ROT4 may also mediate the trade-offs between immunity and development. Given that hundreds of proteins were identified as ROT4-interacting candidates in our mass spectral analysis, new mechanisms may be further characterized. Furthermore, S-acylation is reversible in plant cells (Zheng et al, 2019), and the identification of enzymes controlling the S-acylation status of ROT4 can uncover its dynamic regulation. Collectively, our current work shows that the S-acylation of ROT4 connects non-secreted and secreted protein signaling pathways in plant immunity. This will improve our understanding of the function of plant small peptide modifications and will aid in developing disease resistance in crops.

## Methods

### Plant materials and growth conditions

The seeds of *Arabidopsis thaliana* (Col-0) were surface sterilized and plated on Murashige and Skoog medium containing 1.5% sucrose and 0.7% (w/v) agar in the dark for 2 d and subsequently moved to grow at 22 °C under 16 h of light and 8 h of darkness. The seedlings were transferred to the soil one week after germination. The *pepr1-1* (SALK_059281) and *pepr1-2* (SALK_014538) seeds were described previously (Yamaguchi et al, 2010).

### Construction of plasmids and transgenic plants

For the overexpression of *GFP-ROT4* in protoplasts and transgenic plants, the CDS of *ROT4* was cloned into a pCAMBIA-*35S:GFP* vector. The mutated version of *ROT4* was generated via site-directed mutagenesis. *RTFL13* and *RTFL18* were also cloned into pCAMBIA-*35S:GFP* for the expression of *GFP-RTFL13* and *GFP-RTFL18*, respectively. The sequence information of primers used in this study is included in Appendix Table S1. The generation of transgenic Arabidopsis plants was performed via the floral dipping method mediated by Agrobacterium. The T3 homozygous offspring of at least two independent lines with similar *ROT4* overexpression levels were selected for further analysis.

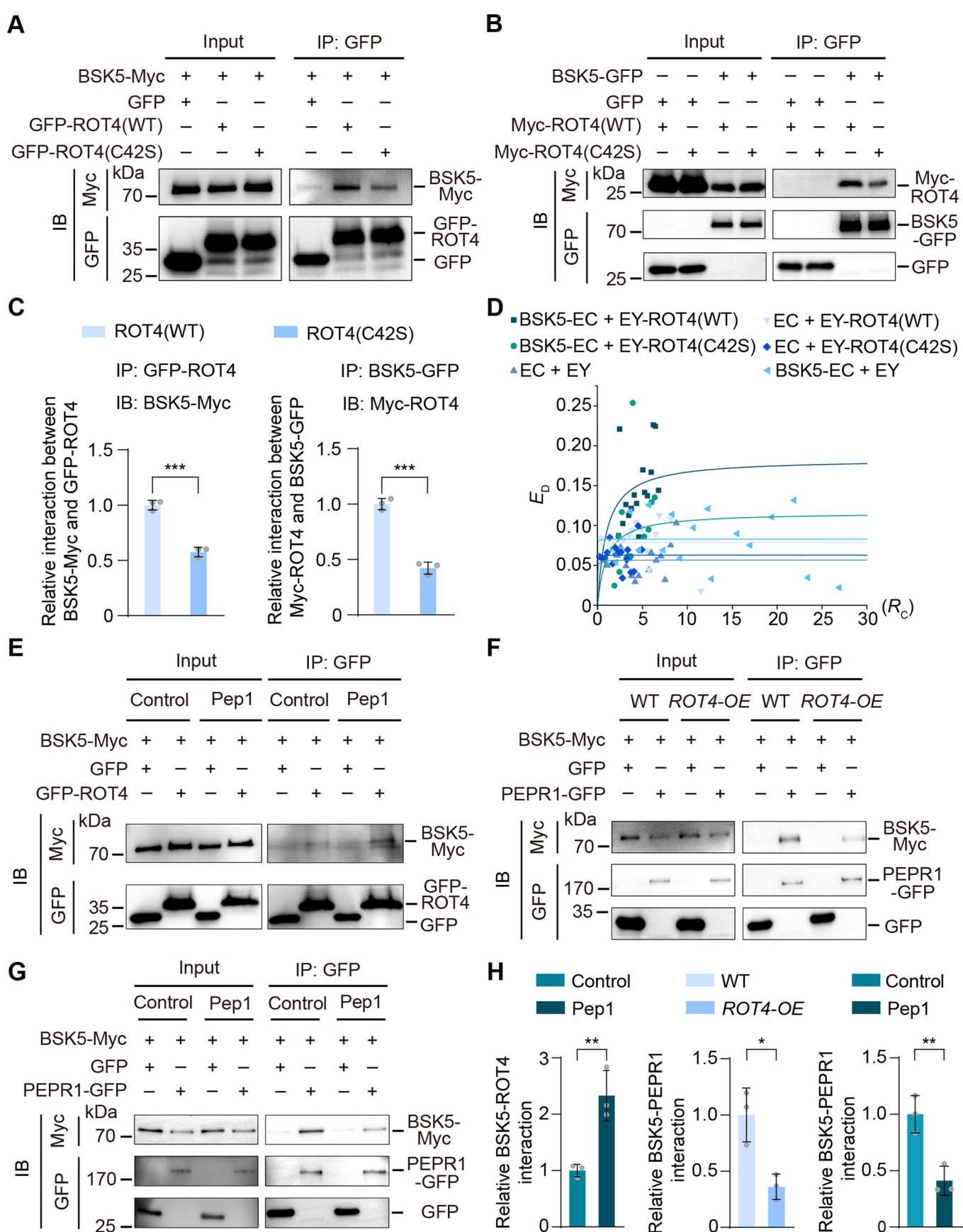

◀ **Figure 3. S-acylation of ROT4 enhances its association with BSK5 for immune activation.**

(A–C) Effect of S-acylation of ROT4 on its interaction with BSK5 using co-IP assays. In Arabidopsis protoplasts, BSK5-Myc was co-expressed with GFP, GFP-ROT4(WT), or GFP-ROT4(C42S) in (A); GFP or BSK5-GFP was co-expressed with Myc-ROT4(WT) or Myc-ROT4(C42S) in (B). Co-IP was performed using anti-GFP Agarose and the representative immunoblots with anti-GFP or anti-Myc antibodies from three biologically independent experiments are shown (A, B). Immunoblot signals were quantified by ImageJ and the interaction intensity was calculated from relative signal ratios (pulldown/input). The relative interaction intensity in the ROT4(WT) samples was set to 1. The quantitative data (C) are representative of three biologically independent experiments. (D) Detection of the effect of ROT4 S-acylation on its interaction with BSK5 in a FRET assay. FRET efficiency ($E_D$) values were distributed into bins of different sizes according to the $R_C$ (acceptor-donor concentration ratio) and plotted against the $R_C$. For FRET efficiency quantification, 16 individual protoplasts in each sample were used and the saturation binding curves are shown. Patterns were consistent in three biologically independent experiments. (E) Effect of Pep1 on the interaction between ROT4 and BSK5. BSK5-Myc was co-expressed with GFP-ROT4 or GFP in protoplasts. Twenty-four hours after transfection, the cells were treated with or without 1 μM of Pep1 for 20 min then were collected for co-IP using anti-GFP agarose. The representative immunoblots from three biologically independent experiments are shown. (F, G) Effect of *ROT4* overexpression and Pep1 treatment on the interaction between BSK5 and PEPR1. GFP or PEPR1-GFP was co-expressed with BSK5-Myc in protoplasts generated from wild-type or *ROT4*-overexpressing plants (F); GFP or PEPR1-GFP was co-expressed with BSK5-Myc in wild-type protoplasts with or without 1 μM of Pep1 treatment for 20 min (G). Co-IP was performed using anti-GFP agarose and the representative immunoblots with anti-GFP or anti-Myc antibodies from three biologically independent experiments are shown. (H) The quantitative analysis of the interaction in (E–G). Left: effect of Pep1 on the BSK5-ROT4 interaction from (E); Middle: effect of *ROT4* overexpression on the BSK5-PEPR1 interaction from (F); Right: effect of Pep1 on the BSK5-PEPR1 interaction from (G). Immunoblot signals were quantified by ImageJ and the interaction intensity was calculated from relative signal ratios (pulldown/input) of BSK5-Myc. The relative interaction intensity in the WT and control samples was set to 1. The data are from three biologically independent experiments. Data information: In (C, H), data are presented as mean ± SD. *$P < 0.05$, **$P < 0.01$, ***$P < 0.001$, Student's *t*-test (two-tailed). Source data are available online for this figure.

## Biotin-switch assay

The biotin-switch assay was conducted as previously described (Liu et al, 2023). At 24 h after protoplast transfection, the proteins were extracted in lysis buffer (100 mM HEPES pH 7.5, 1 mM EDTA, 2 mM TCEP [Tris(2-carboxyethyl)phosphine], 0.1% SDS, and 1 × protease inhibitor cocktail) and incubated at 50 °C for 5 min. After mixing with an equal volume of blocking buffer (100 mM HEPES pH 7.5, 1 mM EDTA, 5% SDS, and 0.5% MMTS [S-Methyl methanethiosulfonate]), the samples were incubated at 40 °C for 10 min. Three volumes of acetone were added for protein precipitation overnight at −20 °C. The mixtures were then spun at 5000 × g for 10 min. The pellets in the tubes were rinsed with 70% acetone and resuspended in 200 μL of resuspension buffer (1 × PBS pH 7.4, 2% SDS, and 8 M urea). The 200 μL of suspension was divided into two tubes; the sample in each tube was incubated for 1 h with a mixture including 50 μL of 4 mM biotin-HPDP (N-[6-(biotinamido)hexyl]-3′-(2′-pyridyldithio)propionamide), 2 μL of 100 mM EDTA, and 1 μL of 100 × protease inhibitor cocktail, and either 50 μL of 2 M NH₂OH (pH 7.4) or Tris-HCl (pH 7.4). Proteins were collected by methanol-chloroform precipitation and dissolved in 100 μL of resuspension buffer. A total of 20 μL of suspension was saved as the input and the rest was mixed with 720 μL of PBS, including 0.2% of Triton X-100, and incubated with Streptavidin-Agarose for 1.5 h. The agarose was rinsed with wash buffer (1 × PBS pH 7.4, 500 mM NaCl, and 0.1% SDS) and 1 × PBS (pH 7.4). Following this, 70 μL of wash buffer containing 5% β-mercaptoethanol was mixed with the Agarose for 20 min to elute S-acylated proteins for SDS–PAGE and immunoblotting using an anti-GFP antibody (TransGen Biotech, HT801-01).

## Confocal microscopy

To detect the subcellular localization of proteins in protoplasts, fluorescent signals were detected 24 h after the transfection of protoplasts generated from 3-week-old Arabidopsis rosette leaves. To measure subcellular localization in intact cells of transgenic plants, the roots or cotyledons of 6-day-old seedlings were used. All subcellular localization images were obtained under a laser-scanning confocal microscope Zeiss LSM 800.

## Cell fractionation

The assay was conducted following a previously described method (Liu et al, 2023). Protein lysates were prepared in homogenization buffer (50 mM Tris-HCl, pH 7.4, 13% sucrose, 1 mM EDTA, 150 mM NaCl, 1 × protease inhibitor cocktail) and incubated for 30 min at 4 °C. The lysates were spun twice at 6000 × g at 4 °C for 10 min and the supernatants were subjected to ultra-centrifugation at 80,000 × g at 4 °C for 1 h for division into pellet and soluble fractions. The pellet fraction was dissolved in the same volume of homogenization buffer. The proteins in the pellet and soluble fractions were analyzed by SDS–PAGE and immunoblotting.

## RNA-Seq

RNA from 3-week-old Arabidopsis plants was prepared using the RNAprep Pure Plant Plus Kit (Tiangen) for RNA-Seq analysis. RNA quality was measured via an Agilent 2100 Bioanalyzer and mRNA was purified to construct sequencing libraries. The libraries from three biologically independent replicates were sequenced on the Illumina HiSeq 2500 platform at Novogene. Raw sequencing reads were filtered with Trimmomatic and the clean reads were mapped to the Arabidopsis genome (TAIR10). Fragments per kilobase of exon model per million were identified using Cufflinks. The DESeq2 package in R was used to analyze the DEGs (FDR < 0.01, |log₂ fold change| > 1). GO and KEGG enrichment were performed using the clusterProfiler package in R ($P < 0.05$).

## Quantitative RT-PCR

RNA was prepared via the RNAprep Pure Kit (Magen) with DNase I for reverse transcription using the PrimeScript RT kit (Vazyme). Quantitative PCR was performed with SYBR qPCR Mix (Tsingke) in the CFX 96 C1000 Thermal Cycler (Bio-Rad).

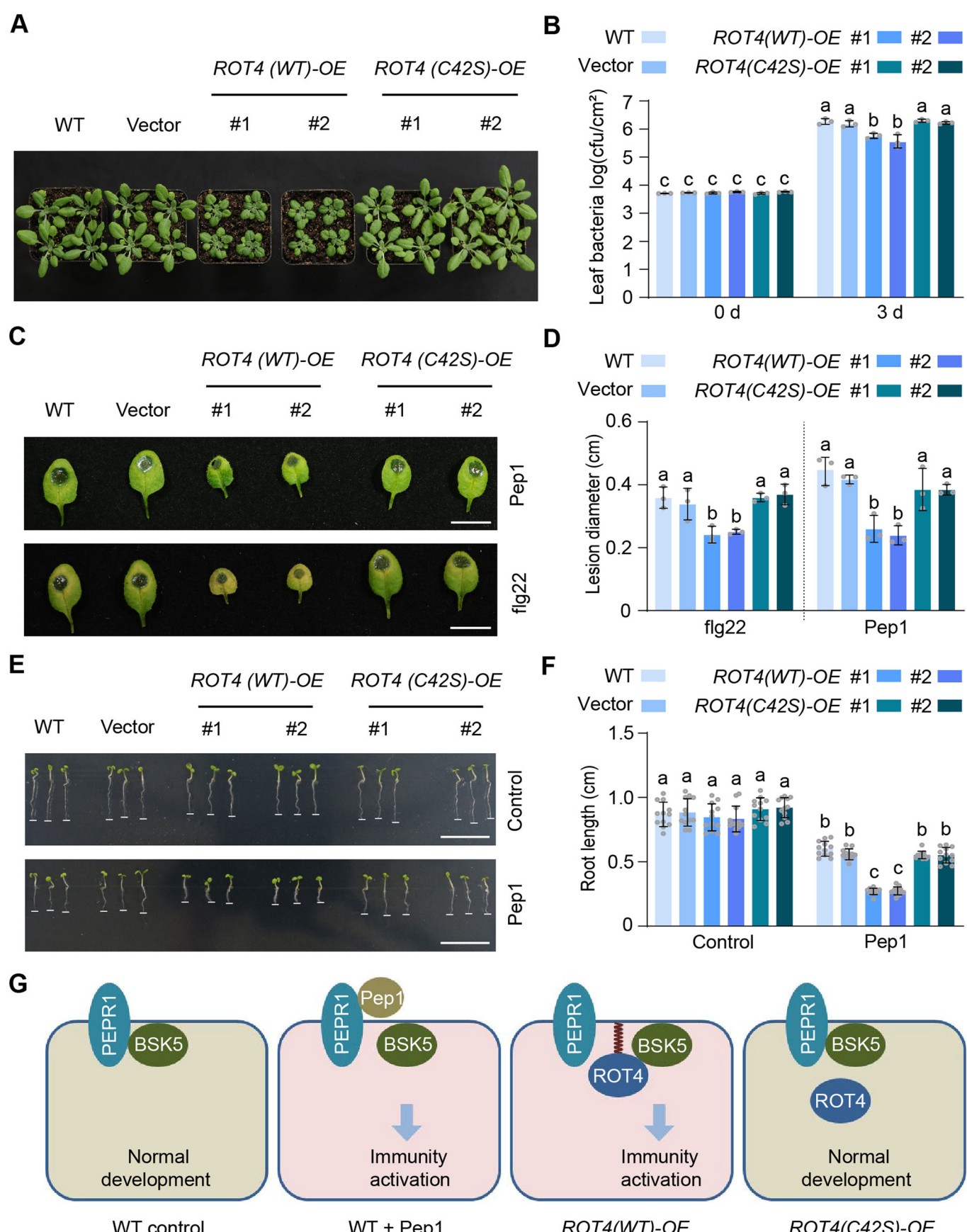

**Figure 4. S-acylation is essential for the function of ROT4 in plant immunity and pathogen resistance.**

(A) Developmental phenotypes of the wild-type and C42S versions of *GFP-ROT4* overexpressing plants. Non-transgenic wild-type plants and GFP vector transgenic plants were included as controls. The 3-week-old plants were photographed and the representative image from three biologically independent experiments is shown. (B) Effect of S-acylation on the function of ROT4 in the resistance to pathogenic bacteria. The indicated 3-week-old plants were inoculated with *Pseudomonas syringae Pst* DC3000. The same sizes of leaf areas ($n \geq 3$ in each sample) were collected at 0 d and 3 d after inoculation. The data of colony numbers in the unit area are from three biologically independent experiments. (C, D). Effect of S-acylation defect on the function of ROT4 in the resistance to pathogenic fungi. The indicated 3-week-old plants were treated with 200 nM of flg22 or 100 nM of Pep1 for 24 h. The leaves were detached and further inoculated with *Botrytis cinerea*. The symptoms were observed 3 d after inoculation and the representative images are included in (C). The quantitative analysis of lesion diameters from three biologically independent experiments is shown in (D). (E, F) Root development in response to Pep1 in the wild-type or C42S version of *ROT4* overexpressing seedlings. The seeds were germinated on the medium with 0 or 20 nM of Pep1. The phenotypes were recorded 5 d after germination (E). The statistical data (F) are mean ± SD from 12 roots in an experiment and three biologically independent experiments displayed similar patterns. (G) Proposed model for the function of S-acylation of ROT4 in plant immunity activation. Under normal conditions, PEPR1 interacts with BSK5; Pep1 binds to PEPR1 and enhances the dissociation of PEPR1-BSK5 for immune activation; an excess amount of S-acylated ROT4 interacts with BSK5 to enhance its dissociation with PEPR1; and S-acylation is essential for the function of ROT4 in the activation of plant immunity. Data information: In (B, D, F), data are presented as mean ± SD; significance analysis was performed using one-way ANOVA followed by Tukey's multiple comparison tests ($P < 0.05$). Source data are available online for this figure.

## IP for mass spectral analysis

Ten-day-old *GFP* and *GFP-ROT4* overexpressing seedlings (two biologically independent replicates) were ground in liquid nitrogen and mixed with 1 mL IP buffer (50 mM Tris-HCl pH 7.4, 150 mM NaCl, 1 mM MgCl$_2$, 10% glycerol, 0.5% Nonidet P-40, 1 × protease inhibitor cocktail, and 25 µM MG132) for incubation at 4 °C for 1 h. After centrifugation at 18000 g at 4 °C for 20 min, the supernatant was incubated with prewashed anti-GFP Agarose at 4 °C for 2 h. The beads were then washed three times with wash buffer (50 mM Tris-HCl pH 7.4, 150 mM NaCl, 1 mM MgCl$_2$, 10% glycerol, and 0.02% Nonidet P-40). The precipitated proteins were eluted by incubating the beads with 70 µL of 0.2 M glycine (pH 2.5) for 10 min. The eluted sample was mixed with a half volume of Tris buffer (1 M; pH 10.4). The IP quality was confirmed by silver staining using Protein Stains K (Sangon Biotech) and the samples were sent to the Wininnovate Bio Company (Shenzhen, China) for mass spectral analysis.

## Mass spectral analysis

The mass spectral analysis was performed according to the following standard protocol from the Wininnovate Bio Company (Shenzhen, China).

Proteins were mixed with 200 µL of 8 M urea in a Nanosep centrifugal device (Pall, USA) for further centrifugation at 12,000 × g for 20 min. A total of 200 µL of 8 M urea with 10 mM DTT was then added and incubated at 37 °C for 2 h. After centrifugation, 200 µL of 8 M urea solution with 50 mM iodoacetamide was added to the filter tube for incubation for 15 min in the dark. The filter tube was then washed three times with 200 µL of 8 M urea and 200 µL of 25 mM ammonium bicarbonate via centrifugation. Following this, 100 µL of 25 mM ammonium bicarbonate with 0.01 µg/µL trypsin was mixed with the sample in the tubes and incubated at 37 °C for 12 h. The peptide fragments in the tubes were eluted twice with 100 µL of 25 mM ammonium bicarbonate via centrifugation and the flow-through fractions were lyophilized.

The lyophilized peptides were dissolved in water with 0.1% formic acid and 2 µL of aliquots were loaded to a nanoViper C18 (Acclaim PepMap 100, 75 µm × 2 cm) trap column. Chromatography separation was conducted on the Easy nLC 1200 system

(ThermoFisher, USA). Trapping and desalting were performed with 20 µL of 100% solvent A (0.1% formic acid). Elution gradients of 5–38% solvent B (0.1% formic acid, 80% acetonitrile) in 60 min were used on an analytical column (Acclaim PepMap RSLC, 75 µm × 25 cm C18-2 µm 100 Å). Data-dependent acquisition mass spectral techniques were applied to obtain tandem MS information on a Q Exactive mass spectrometer (ThermoFisher, USA) fitted by a Nano Flex ion source. Data were acquired using an ion spray voltage of 1.9 kV and an interface heater temperature of 275 °C. The MS/MS data were analyzed using PEAKS Studio 8.5. The local false discovery rate at PSM was 1% after searching in the *Arabidopsis thaliana* database with a maximum of two missed cleavages. The precursor and fragment mass tolerance were set to 10 ppm and 0.05 Da, respectively.

## Co-IP

*BSK5* was cloned into the pBA002-*35S:Myc$_5$* or pCAMBIA1300-*UBQ:GFP* vector; the wild-type and C42S version of *ROT4* was cloned into the pCAMBIA-*35S:GFP* or pCanG-*35S:Myc$_5$* vector; and *PEPR1* was cloned into the pCAMBIA1300-*UBQ:GFP* vector. The indicated plasmid pairs were transformed into Arabidopsis protoplasts. After 24 h, the protoplasts were collected for protein extraction in 700 µL of lysis buffer (50 mM Tris-HCl pH 7.4, 150 mM NaCl, 1 mM MgCl$_2$, 10% glycerol, 0.5% Nonidet P-40, 1 × protease inhibitor cocktail, and 25 µM MG132) via 30 min of incubation. The lysates were centrifuged at 18,000 × g and the supernatants were incubated with prewashed Anti-GFP Nanobody Agarose Beads (AlpaLife) at 4 °C for 2 h. The beads were rinsed three times with wash buffer (50 mM Tris-HCl pH 7.4, 150 mM NaCl, 1 mM MgCl$_2$, 10% glycerol, and 0.02% Nonidet P-40). Following this, 70 µL of 2 × protein sample buffer was added to the beads and boiled for 6 min. The protein samples were used for SDS–PAGE and immunoblotting with anti-MYC (TransGen Biotech, HT101-01) and anti-GFP (TransGen Biotech, HT801-01) antibodies.

## FRET

FRET analysis was performed on an automated multimodal quantitative FRET microscopic imaging system (Auto-MS). The system consists of a wide-field fluorescence microscope (IX73,

Olympus, Japan) equipped with a metal halide lamp (HGLGPS, Olympus, Japan), an objective (60× /1.42 NA oil, Olympus, Japan), and a CMOS (ORCA-Flash4.0, Hamamatsu, Japan). FRET efficiency is quantitatively measured using 3-cube-based FRET microscopy (E-FRET) proposed by (Sun et al, 2019). The saturation binding curve was fitted according to the function $E_D = E_D max \ R_C/$ $R_C/(Kd + R_C)$ (Aranovich et al, 2012). EYFP-ROT4(WT) or EYFP-ROT4-C42S was co-expressed with BSK5-ECFP for FRET measurements. Free ECFP and EYFP were co-expressed as a negative control.

## BiFC

The wild-type or C42S version of *ROT4* was cloned into the pSAT6-*nEYFP* vector, and *BSK5* was cloned into the pSAT6-*cEYFP* vector. The plasmid pairs were transformed into Arabidopsis protoplasts; 24 h after transfection, YFP fluorescence was detected under a Zeiss LSM 800 laser-scanning confocal microscope.

## Pathogen inoculation

For the inoculation of *Pseudomonas syringae* pv. *Tomato* DC3000, $5 \times 10^8$ CFU/mL of bacterial suspension in 10 mM $MgSO_4$ with 0.03% Silwet L-77 was used to spray 3-week-old Arabidopsis plants and the inoculated plants were kept at high humidity for 1 day. Leaf bacterial number in the same size of areas ($0.13 \ cm^2$) was detected at the indicated time points after inoculation.

For the inoculation of *Botrytis cinerea*, rosette leaves from 3-week-old plants were treated with 200 nM flg22 or 100 nM Pep1 for 24 h. Following this, 5 μL of spore suspension ($3 \times 10^6$ spores/mL) was deposited on the detached leaves in a plate with 1% agar. The symptoms were recorded after incubating at 25 °C for 3 d.

## Accession numbers

The accession numbers of genes in this study are as follows: *ROT4* (AT2G36985), *RTFL13* (AT3G23635), *RTFL18* (AT5G16023), *BSK5* (AT5G59010), *PEPR1* (AT1G73080), *PR1* (AT2G14610), *FRK1* (AT2G19190), *EDS1* (AT3G48090), *WRKY25* (AT2G30250), *CML37* (AT5G42380), *CML41* (AT3G50770), *CML47* (AT3G47480), *CNGC10* (AT1G01340), *PAT12* (AT4G00840).

## Data availability

The RNA-Seq datasets were deposited in the BioProject: accession PRJNA979946. The datasets produced in this study are available in the following databases: RNA-Seq data: National Center for Biotechnology Information BioProject data (https://www.ncbi.nlm.nih.gov/bioproject/PRJNA979946). SRA names: SRR24825716, SRR24825715, SRR24825714, SRR24825713, SRR24825712, and SRR24825711.

## Peer review information

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

## Acknowledgements

We would like to thank the Nottingham Arabidopsis Stock Centre for the *PEPR1* mutant seeds and Prof. Nan Yao and Prof. Lijing Liu for providing bacterial strains. This work was supported by Major Program of Guangdong Basic and Applied Research (2019B030302006), National Natural Science Foundation of China (32270752, 31970531, 32270292), Natural Science Foundation of Guangdong (2021A1515011151, 2019A1515110330, 2018B030308002), Guangdong Modern Agro-industry Technology Research System (2023KJ114), the Program for Changjiang Scholars, and the Guangdong Special Support Program of Young Top-Notch Talent in Science and Technology Innovation (2019TQ05N651).

## Author contributions

**Wenliang Li**: Investigation; Writing—original draft. **Tushu Ye**: Investigation. **Weixian Ye**: Investigation. **Jieyi Liang**: Investigation. **Wen Wang**: Methodology. **Danlu Han**: Investigation. **Xiaoshi Liu**: Investigation. **Liting Huang**: Investigation. **Youwei Ouyang**: Investigation. **Jianwei Liao**: Investigation. **Tongsheng Chen**: Methodology. **Chengwei Yang**: Supervision; Funding acquisition. **Jianbin Lai**: Supervision; Funding acquisition; Writing—original draft; Writing—review and editing.

## Disclosure and competing interests statement

The authors declare no competing interests.

# Expanded View Figures

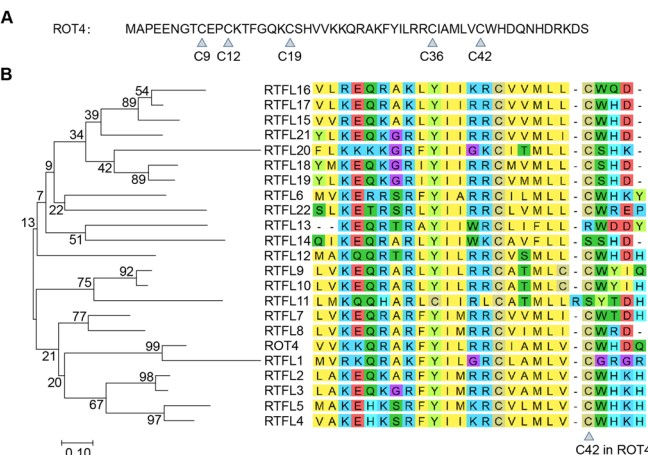

**A**

ROT4:  MAPEENGTCEPCKTFGQKCSHVVKKQRAKFYILRRCIAMLVCWHDQNHDRKDS
       C9 C12    C19           C36   C42

**B**

**Figure EV1. Conservation of S-acylation sites in Arabidopsis RTFL members.**

(A) The positions of potential S-acylation sites on ROT4. The cysteine residues are indicated on the whole protein of ROT4. (B) Protein alignments of the conserved domain of Arabidopsis RTFL members. The sequences of the Arabidopsis RTFL proteins were aligned using ClustalW. The conserved domain is shown and the C42 residue in ROT4 is indicated. The phylogenetic tree was constructed using full-length protein sequences with a neighbor-joining method (bootstrap, 1000 replicates).

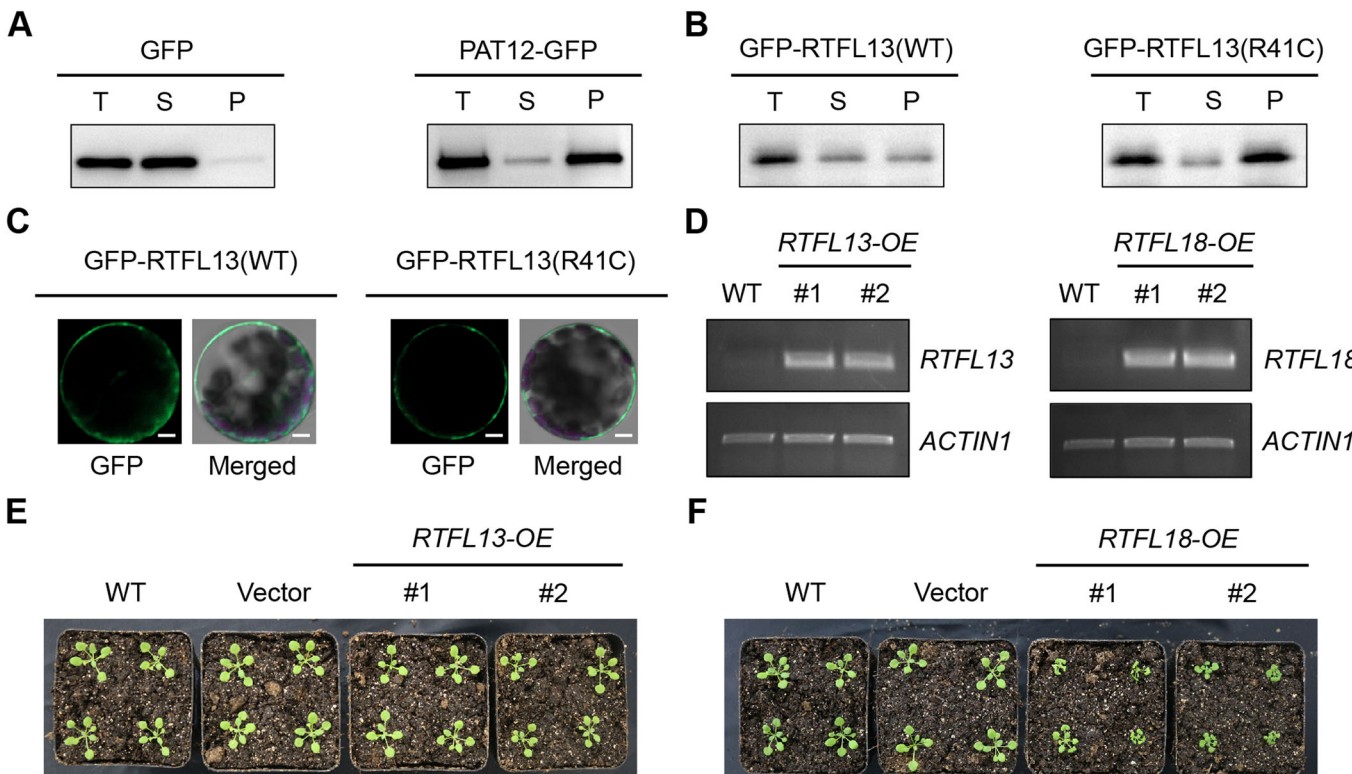

**Figure EV2.  Contribution of S-acylation to the function of RTFL proteins.**

(**A**) Specificity verification of the cell fractionation assay. Free GFP (in cytosol fraction) or PAT12-GFP (in membrane fraction) was expressed in protoplasts. Total proteins (T) from protoplasts were divided into pellet (P) and soluble (S) fractions via ultra-centrifugation. The representative anti-GFP immunoblot from three biologically independent experiments is shown. (**B, C**) Effect of an R-to-C substitution on the subcellular localization of RTFL13. The WT or R41C version of GFP-RTFL13 was expressed in protoplasts. The representative immunoblotting results of cell fractionation from three biologically independent experiments are shown in (**B**). The representative GFP (green) and merged (with the bright field in gray and chloroplast auto-fluorescence in magenta) signals from three biologically independent experiments are shown in (**C**). Scale bars: 5 µm. (**D–F**) The phenotypes of *RTFL13* and *RTFL18* overexpressing plants. The expression level of *RTFL13* or *RTFL18* overexpressing lines was verified by RT-PCR (**D**). The RT-PCR data are representative of three biologically independent experiments. *ACTIN1* was used as an internal control. The representative phenotypes of 2-week-old WT, vector control, and two independent lines of *RTFL13* or *RTFL18* overexpressing plants from three biologically independent experiments are shown (**E, F**). Source data are available online for this figure.

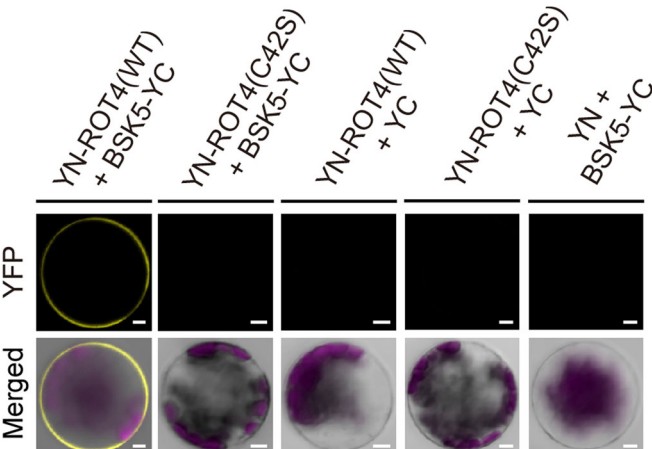

**Figure EV3. Effect of ROT4 S-acylation on its interaction with BSK5 in a BiFC assay.**

The indicated protein pairs were expressed in protoplasts for 24 h before confocal microscopy. YN or YC, the N or C fragment of YFP. The representative YFP (yellow) and Merged (with the bright field in gray and chloroplast autofluorescence in magenta) from three biologically independent experiments are shown. Scale bars: 5 µm. Source data are available online for this figure.

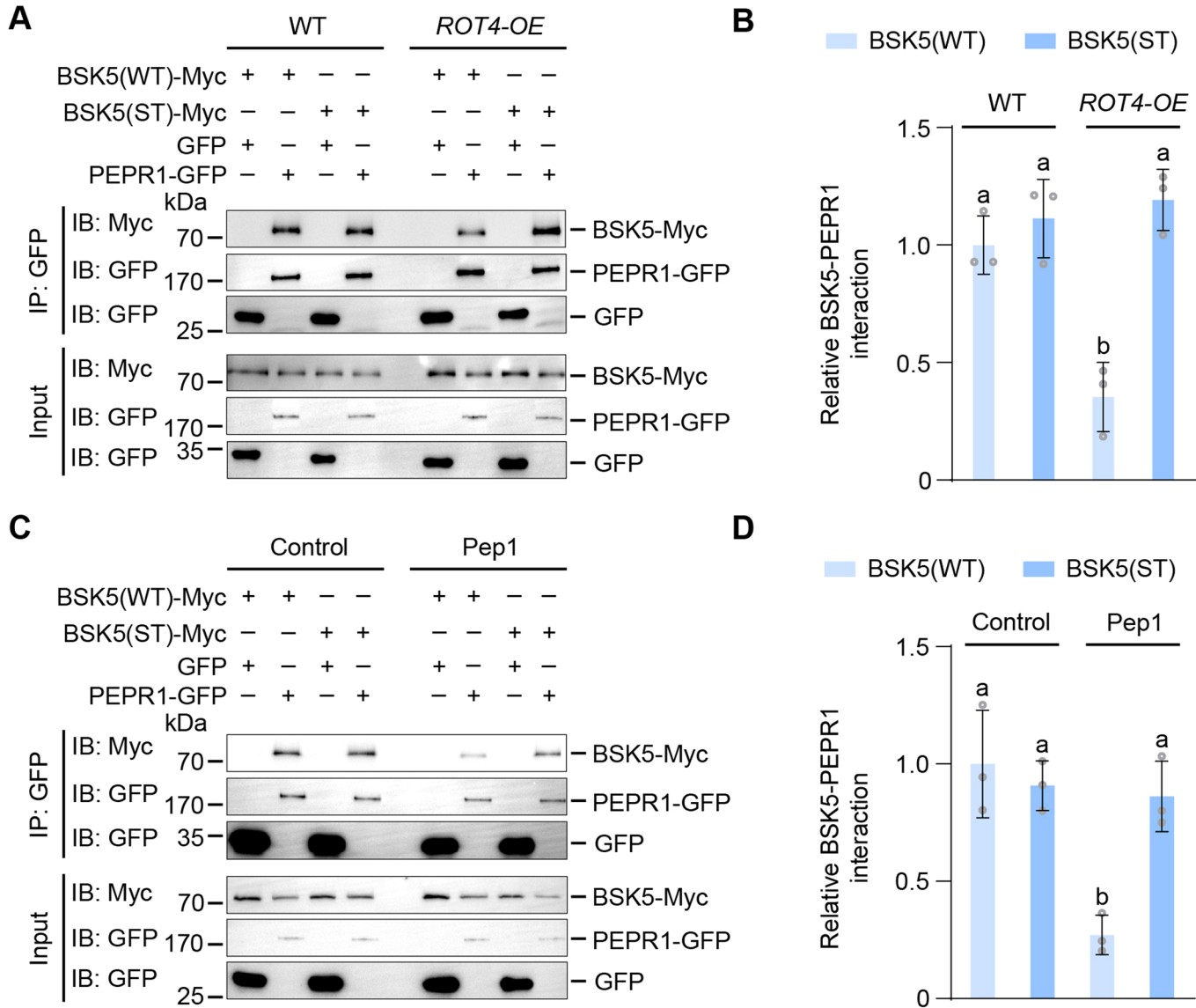

**Figure EV4. Effect of *ROT4* overexpression and Pep1 treatment on the association of PEPR1 and phosphorylation defective BSK5.**

(A, B) Effect of *ROT4* overexpression on the interaction between BSK5(S209A/T210A) and PEPR1. GFP or PEPR1-GFP was co-expressed with BSK5(WT)-Myc or BSK5(S209A/T210A)-Myc in protoplasts generated from wild-type or *ROT4*-overexpressing plants. ST: S209A/T210A. Co-IP was performed using anti-GFP Agarose and the representative immunoblots with anti-GFP or anti-Myc antibodies are shown in (A). The quantitative analysis of the BSK5-PEPR1 interaction from three biologically independent experiments is shown in (B). Immunoblot signals were quantified by ImageJ and the interaction intensity was calculated from relative signal ratios (pulldown/input) of BSK5-Myc. The relative interaction intensity in the sample of BSK5(WT) in the WT cells was set to 1. (C, D) Effect of Pep1 on the interaction between BSK5(S209A/T210A) and PEPR1. GFP or PEPR1-GFP was co-expressed with BSK5(WT)-Myc or BSK5(S209A/T210A)-Myc in wild-type protoplasts with or without 1 μM of Pep1 treatment for 20 min. ST: S209A/T210A. Co-IP was performed using anti-GFP Agarose and the representative immunoblots with anti-GFP or anti-Myc antibodies are shown in (C). The quantitative analysis of the BSK5-PEPR1 interaction from three biologically independent experiments is shown in (D). Immunoblot signals were quantified by ImageJ and the interaction intensity was calculated from relative signal ratios (pulldown/input) of BSK5-Myc. The relative interaction intensity in the sample of BSK5(WT) under a control condition was set to 1. Data information: In (B, D), data are presented as mean ± SD; significance analysis was performed using one-way ANOVA followed by Tukey's multiple comparison tests ($P < 0.05$). Source data are available online for this figure.

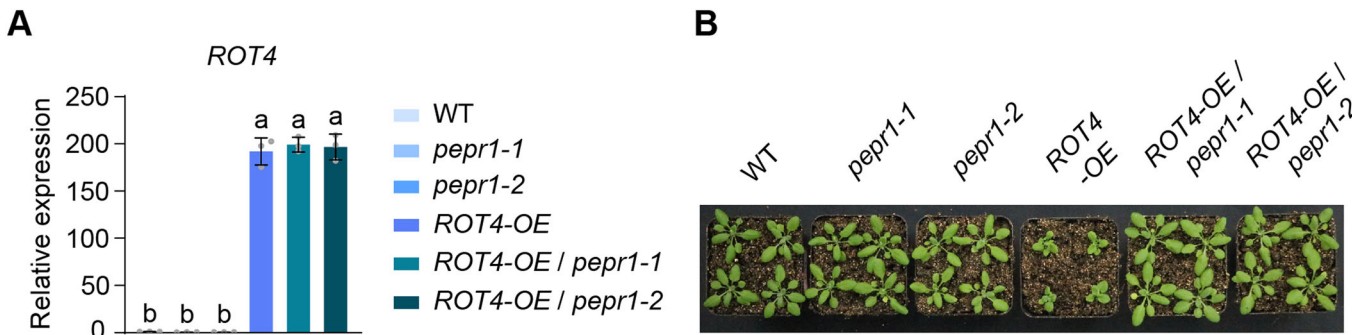

**Figure EV5. Effect of *ROT4* overexpression on development in the *PEPR1* mutant background.**

(A) Verification of the transgenic plants by quantitative RT-PCR. *ROT4* was overexpressed in the WT, *pepr1-1*, or *pepr1-2* mutant background and the transgenic lines were verified by quantitative RT-PCR. *ACTIN2* was used as an internal control. The relative expression level of *ROT4* in the WT plants was set to 1. The quantitative RT-PCR data are from triplicated technological repeats, three biologically independent experiments showed similar patterns. (B) Developmental phenotypes of the indicated plants. Non-transgenic WT and *PEPR1* mutants are included as controls. The 3-week-old plants were photographed. The representative image of plant phenotypes from three biologically independent experiments is shown. Data information: In (A), data are presented as mean ± SD; significance analysis was performed using one-way ANOVA followed by Tukey's multiple comparison tests ($P < 0.05$). Source data are available online for this figure.

