## [Peer Review File · EMBO Reports]

S-acylation of a non-secreted peptide controls plant immunity via secreted-peptide signal activation

Wenliang Li, Tushu Ye, Weixian Ye, Jieyi Liang, Wen Wang, Danlu Han, Xiaoshi Liu, Liting Huang, Youwei Ouyang, Jianwei Liao, Tongsheng Chen, Chengwei Yang, and Jianbin Lai

DOI: [10.15252/embr.202357634](https://doi.org/10.15252/embr.202357634)

Corresponding author(s): Jianbin Lai (20141062@m.scnu.edu.cn) , Chengwei Yang (yangchw@scnu.edu.cn)

Review Timeline:

Submission Date:	12th Jun 23
Editorial Decision:	1st Aug 23
Revision Received:	25th Oct 23
Editorial Decision:	21st Nov 23
Revision Received:	25th Nov 23
Accepted:	30th Nov 23

Editor: Achim Breiling

Transaction Report:

Dear Prof. Lai,

Thank you for the submission of your manuscript to EMBO reports. I have now received the reports from the three referees that were asked to evaluate your study, which can be found at the end of this message.

As you will see, the referees think that these findings are of high interest. However, they have several comments, concerns, and suggestions to improve the manuscript, indicating that a major revision is necessary to allow publication of the study in EMBO reports. As the reports are below, and all the referee concerns need to be addressed, I will not detail them here.

Given the constructive referee comments, I would like to invite you to revise your manuscript with the understanding that all referee concerns must be addressed in the revised manuscript and in a detailed point-by-point response. Acceptance of your manuscript will depend on a positive outcome of a second round of review. It is EMBO reports policy to allow a single round of revision only and acceptance of the manuscript will therefore depend on the completeness of your responses included in the next, final version of the manuscript.

- 1) a .docx formatted version of the final manuscript text (including legends for main figures, EV figures and tables), but without the figures included. Figure legends should be compiled at the end of the manuscript text.
- 2) individual production quality figure files as .eps, .tif, .jpg (one file per figure), of main figures (up to 8) and EV figures (up to 5). Please upload these as separate, individual files upon re-submission.

- 4) a complete author checklist, which you can download from our author guidelines (<https://www.embopress.org/page/journal/14693178/authorguide>). Please insert page numbers in the checklist to indicate where the requested information can be found in the manuscript. The completed author checklist will also be part of the RPF.

- 5) that primary datasets produced in this study (e.g. RNA-seq, ChIP-seq, structural and array data) are deposited in an

appropriate public database. If no primary datasets have been deposited, please also state this in a dedicated section (e.g. 'No primary datasets have been generated and deposited'), see below.

The accession numbers and database should be listed in a formal "Data Availability" section (placed after Materials & Methods) that follows the model below. This is now mandatory (like the COI statement). Please note that the Data Availability Section is restricted to new primary data that are part of this study. This section is mandatory. As indicated above, if no primary datasets have been deposited, please state this in this section

Data availability

8) Regarding data quantification and statistics, please make sure that the number "n" for how many independent experiments were performed, their nature (biological versus technical replicates), the bars and error bars (e.g. SEM, SD) and the test used to calculate p-values is indicated in the respective figure legends (also for potential EV figures and all those in the final Appendix). Please also check that all the p-values are explained in the legend, and that these fit to those shown in the figure. Please provide statistical testing where applicable. Please avoid the phrase 'independent experiment', but clearly state if these were biological or technical replicates. Please also indicate (e.g. with n.s.) if testing was performed, but the differences are not significant. In case n=2, please show the data as separate datapoints without error bars and statistics. See also: <http://www.embopress.org/page/journal/14693178/authorguide#statisticalanalysis>

9) Please add scale bars of similar style and thickness to microscopic images, using clearly visible black or white bars (depending on the background). Please place these in the lower right corner of the images themselves. Please do not write on or near the bars in the image but define the size in the respective figure legend.

10) Please also note our reference format:

12) We now use CRedit to specify the contributions of each author in the journal submission system. CRedit replaces the author contribution section. Please use the free text box to provide more detailed descriptions and do not provide your final manuscript text file with an author contributions section. See also our guide to authors: <https://www.embopress.org/page/journal/14693178/authorguide#authorshipguidelines>

13) We would encourage you to use 'Structured Methods', our new Materials and Methods format. According to this format, the

Materials and Methods section should include a Reagents and Tools Table (listing key reagents, experimental models, software and relevant equipment and including their sources and relevant identifiers) followed by a Methods and Protocols section in which we encourage the authors to describe their methods using a step-by-step protocol format with bullet points, to facilitate the adoption of the methodologies across labs. More information on how to adhere to this format as well as downloadable templates (.doc or .xls) for the Reagents and Tools Table can be found in our author guidelines (section 'Structured Methods'):

14) Please order the manuscript sections like this, using these names:

Title page - Abstract - Keywords - Introduction - Results - Discussion - Materials and Methods - Data availability section - Acknowledgements - Disclosure and Competing Interests Statement - References - Figure legends - Expanded View Figure legends

15) Please have your revised manuscript carefully proofread by a native speaker.

I look forward to seeing a revised version of your manuscript when it is ready. Please let me know if you have questions or comments regarding the revision.

Yours sincerely,

Referee #1:

Li et al. show that plasma membrane targeting of ROT4 is mediated by S-acylation at Cys42. They further show that overexpression of ROT4, but not of a ROT4 C42S mutant, causes massive upregulation of defense-related genes leading to enhanced resistance against bacterial (*P. syringae* DC3000) and fungal (*B. cinerea*) pathogens. Co-IP mass spec analysis identified the cytoplasmic receptor-like kinase BSK5 as an interactor of ROT4, and the interaction of ROT4 with BSK5 was found to be reduced in the C42S mutant. Further Co-IP experiments suggested that ROT4 attenuates the interaction between BSK5 and the pattern recognition receptor PEPR1. Based on these findings Li et al. propose a model of the mechanism underlying ROT4-dependent induction of immunity. Previous studies showed that the BSK5/PEPR1 complex dissociates upon ligand binding, resulting in the release of BSK5 and pattern-triggered immunity. According to the proposed model, ROT4 facilitates BSK5 release thereby promoting immune signaling.

The paper reveals a novel function of ROT4 in immune signaling, which relies on post-translational modification by S-acylation for targeting to the plasma membrane. ROT4 is proposed to facilitate the release of BSK5 from the pre-existing BSK5/PEPR1 complex, thereby activating pattern-triggered immunity. This is certainly a very interesting hypothesis. The topic is of broad interest and well within the scope of EMBO Reports. The data related to ROT4 localization, ROT4-dependent induction of defense, and S-acetylation at Cys42 as a prerequisite for both, are solid and convincing. However, further evidence is needed to substantiate the model proposed for the ROT4 mode of action.

Major:

1. The authors state that ROT4-mediated 'dissociation of PEPR1 and BSK5 ... mimics the process in the presence of Pep1, activating downstream immunity pathways' (line 201-203). In my view, this is only partially true. In presence of Pep1, BSK5 is not only released but also phosphorylated by PEPR1 in the kinase activation loop. Phosphorylation and kinase activity were shown to be critical for the function of BSK5 in pattern-triggered immunity (Majhi et al. 2019). Therefore, the model implies that BSK5 is phosphorylated in the presence of an excess of ROT4. If this is the case, evidence for BSK5 phosphorylation in ROT4 overexpressing lines should be included. On the other hand, if phosphorylation of BSK5 is not required, the ROT4 overexpression phenotype would be expected in the background of a *bsk5* mutant complemented with a phospho-dead BSK5 variant.

2. Most of the results were obtained by overexpressing ROT4, or the ROT4 C42S mutant. I am concerned that the presence of excessively large amounts of ROT4 in the plasma membrane may lead to interactions that do not occur in wild-type plants. Therefore, strong support for the conclusions drawn could be obtained by including a loss-of-function experiment.

Minor:

1. Additional support for the model could be provided by comparing the ROT4-dependent and the BSK5-dependent

transcriptomes - large overlap expected.

2. The title is somewhat enigmatic. I would prefer a title which names the main players/findings
3. The heat map in Figure 2C is said to confirm upregulation of genes. The heat map is just another way of presenting the data, rather than a confirmation of the results
4. The conclusion that 713 proteins '...were specifically associated with GFP-ROT4' (line 168) is not justified. The list of 713 proteins is likely to contain many that do not interact with GFP-ROT4. Many more biological replicates and statistics would be required to justify this conclusion. Aiming at the identification of proteins interacting at the plasma membrane, GFP-ROT4-C42S would have made a better control than GFP alone.
5. Please describe the principle of the biotin-switch assay, either in the text or in the methods section. A few sentences will suffice (similar how you did it in your previous publication, Liu et al. 2023)
6. I do not understand the experiment shown in Figure 4C,D. Why were the peptides added? No peptides were needed in Figure 4B? Why do you get the same effect for flg22 as for Pep1?
7. Hurst et al 2023, Current Biology 33:1588 should be discussed
8. There are many unusual abbreviations: TCEP, MMTS, HPDP, ... please spell out.
9. Fig. 3H is not really convincing. Yes, the band is weaker when both PEPR1-GFP and Pep1 are present, but this is also observed in the input panel.

Referee #2:

In this manuscript entitled S-acylation of a non-secreted peptide controls plant immunity via secreted-peptide signal activation by Wenliang Li, Tushu Ye and colleagues the authors address the importance of acylation as a posttranslational modification of the non-secreted ROTUNDIFOLIA4 (ROT4) peptide in Arabidopsis. The ROT4 peptide belongs to the RTFL/DEVIL (DVL) peptide family that consist of 23 members in Arabidopsis and that has a conserved domain of about 30 amino acids. The authors of this work use CSS-PALM analysis to identify potential sites of S-acylation (cysteine residues) in the conserved domain of several ROT members, there among ROT4. They explore the functional importance of this S-acylation site in ROT4 by several methods and show that S-acylation is important for another member of the RTFL/DEVIL family (RTFL18). From their experiments the authors conclude that the S-acylation is important for membrane localization and for the interaction between ROT4 and the cytoplasmic kinase BSK5. Using overexpressing (OE) lines the authors conclude that ROT4 is important for immunity responses in Arabidopsis by attenuating the dissociation of the immune LRR-RLK receptor PEPR1 and BSK5, a disassociation that is also triggered when PEPR1 binds its peptide partner, the secreted Pep1 peptide. Furthermore, the authors show through pathogen assays that OE ROT4 lines are less susceptible to *Pseudomonas syringae* pv. tomato DC3000 and the fungal pathogen *Botrytis cinerea* in combination with the pathogenic peptide flg22 or Pep1.

The manuscript is clear and concise although it would benefit from some additional language editing. The figures are of good quality and the results are clearly presented. In most cases it is indicated how many independent times the experiments were repeated and proper statistical analysis are performed. Critical references are given, and the M and M section is adequately written.

This manuscript reports two key findings:

- The importance of acylation for the function of at least two RTFL family members and the proposed function of ROT4 in plant immunity.

This manuscript adds novelty to the importance of posttranslational modification of peptides although it is already known that acylation of receptors in plants stabilizes ligand-induced receptor kinase complex formation during immune signaling (<https://www.sciencedirect.com/science/article/pii/S0960982223002385>). The work in this manuscript will therefore be of interest to the plant molecular biology community.

The finding that acylation is important for the membrane localization of ROT4 and for its OE developmental phenotypes is well founded. However, the conclusion drawn on the role of ROT4 in plant immunity requires further experimental work to be conclusive.

Below follows a list of comments and concerns and some suggestions that could improve the manuscript.

Mayor concerns:

- All the transgenic experiments performed for the OE lines are done in the Col-0 WT background in the presence of the endogenous ROT4 gene, so in the lines where conclusions are drawn for the OE of ROT4 with a cysteine to serine substitution the endogenous ROT4 is still active. In my opinion the results would be more convincing if this was done in a rot4 mutant background. For example when comparing the RNA seq data and QPCR data between the two versions of ROT4 OE for the C to S substitution the active ROT4 peptide is still present.
- For the experiment where RTFL13 was tagged with GFP to show "global" expression in the cell and its lack of S-acylation a more convincing experiment for the essential role of this cysteine for S-acylation would be to include results where the Arginine is substituted for a Cysteine to see if this would restore plasma membrane localization and S-acylation. Also, it would be interesting to compare OE phenotypes between RTFL13 and the other members of this family and see if the change of Arginine to Cysteine would affect the phenotypes.
- From the results from the RNA seq experiments it is not clear to me why the authors choose to focus their follow up experiments in immunity and BSK5. From the GO analysis it would appear that OE ROT4 also induces responses compatible with biotic stresses. A better line of reasoning needs to be made. The authors conclude that the reduced growth rate of the OE ROT4 lines are compatible with an elevated and continuous immune response, but there are many other upregulated genes that could affect the growth phenotype. Also there is no inhibition in root length in the ROT4 OE lines unless Pep1 is added while the model proposes that ROT4 on its own, when OE, activates immunity. In general I think the authors need to be more cautious with drawing conclusions on the role of ROT4 regulating plant immunity based on OE experiments. I follow the authors concern with regard to redundancy among the RTFL/DEVIL family members but CRISPR lines where ROT4 (and other members with overlapping expression patterns) are mutated should be investigated for susceptibility to pathogens to draw more convincing conclusions.
- With regards to the RNAseq experiment it is not clear to me why the ROT4 OE substitution line was not included and only used to later check for selected candidate genes by QPCR. This needs to be addressed.
- In line 175 the authors conclude that BSK5 interacts with the membrane localized ROT4, however, apart from the split YFP experiments there is no in vivo evidence for this interaction. Given that ROT4 localizes nicely to the PM in root tissue it should be possible to do FRET or even better FRET-FLIM (for quantification given that the IP data only shows a reduction in interaction with the ROT4 substituted lines) to show in vivo interactions between BSK5 and ROT4 (and reduced interaction in the substituted lines). Ideally since the authors suggest that Pep1 mediates this interaction after binding PEPR1 the in vivo experiments should be done with and without the presence of exogenous Pep1 (again this should be possible to do in root tissue).
- Line 241: The final conclusion of this work where the authors conclude that ROT4 connects secreted and non-secreted signaling in plant immunity is not shown. The authors show that there is less interaction between BSK5 and PEPR1 in the ROT4 OE lines and that there is less interaction between BSK5 and ROT 4 OE (C42S) lines but there is no experiment where the interaction between BSK5 and ROT4 is measured in response to Pep1.

Minor comments:

Line 46: Previous studies showed that activation expression of the member of this family in Arabidopsis, such as ROTUNDIFOLIA4 (ROT4) or DEVIL1 (DVL1)/RTFL18, results in a phenotype with small and round leaves (Narita et al, 2004; Wen et al, 2004). Please explain what you mean by "activation expression"

Line 62: Although PLANT ELICITOR PEPTIDES (PEPs) were expressed without an N terminus signaling motif, the mature forms of these peptides are secreted outside and recognized by PEP RECEPTOR1/2 (Bartels & Boller, 2015). What is the reasoning for stating that the PEPs were expressed without an N terminal signaling peptide?

Line 96: please make an additional supplementary figure where you show the whole ROT4 protein with all the potential cysteine sites that are identified as potential S-acylation sites.

Line 109: The further cell fractionation data indicated that most of the wild-type GFP-ROT4 accumulated.... Please rewrite.

Line 116: Because the Arabidopsis RTFL family contains 23 members, it is interesting to analyze the C42 site of ROT4 is conserved among the RTFL proteins.... Unclear, please rewrite.

Line 137: unknown. To catch the hints on the pathways regulated by ROT4... please rewrite

Line 139: In several figures, there among figure S2 the authors use the term vector. Please specify what is meant with this. I presume that these are lines expressing the vector used to make the OE lines with the GFP tag.

Figure 2C pathogen is misspelled. Also there is some confusion to using the line numbering here instead of indicating that it represents 2 independent experiments since the S2 only shows 2 lines. It is also confusing to refer to figure S2 with regards to the RNAseq experiments since the OE ROT4 substitution lines were not used for RNA seq only to compare QPCR data.

Line 165: Here the authors use the term GFP plants, other places vector. Please be concise in the terminology used to avoid confusion.

Referee #3:

The key finding of this work is that the conserved S-acylation/S-palmitoylation site Cys42 in ROT4, a non-secreted peptide, is required for activation of plant immune responses through interaction with PEPR1, a receptor-like kinase in Arabidopsis. The role of S-acylation in non-secreted peptides has not yet been investigated and this work will therefore be of interest to both the fields of S-acylation and plant immunology.

The data presented in Figure 4 are excellent and highly compelling - These data demonstrate at a whole organism level how loss of the ROT4 Cys42 S-acylation state impacts plant growth, pathogen resistance and root length. However, the biochemical approaches taken in this work to understand this phenotype, as demonstrated in Figures 1-3, will require strengthening prior to publication.

While this research is novel in its findings, linking the activity of secreted and non-secreted peptides by S-acylation to activate plant immunity, there are several concerns that I would like the authors to address prior to publication.

Major concerns:

1. Based on the data in Figure 1A, the authors state that C42 is the predominant ROT4 S-acylation site. The authors do not clarify whether other cysteines are contributing to this S-acylation state. It may be the case that incomplete MMTS blocking, resulting in the bands appearing in the NH₂OH (-) lane, is masking that C42 is the only S-acylation site. Please could the authors present western blots where there are no bands in the NH₂OH (-) lanes. If this is not possible, please can the authors present quantified data of the reduction in S-acylation. This would clarify whether C42 is the only S-acylation site or is the predominant contributor to ROT4 S-acylation state.

If C42 is not the only S-acylation site, the authors may have to consider repeating the experiments presented here with a double cysteine mutant to show how total loss of S-acylation impacts ROT4 function, which I appreciate will be a lot of work.

2. Figure 1C shows ROT4 and ROT4 C42S membrane fractionation by ultracentrifugation, however, no loading controls are presented. PM and cytoplasmic protein contamination markers should also be included. Please could the authors include loading controls and contamination markers for this experiment.

3. The authors state that the ROT4 C42S mutation reduces ROT4-BSK1 interaction. However, the IP data presented in Fig 3D and 3E are currently not sufficiently compelling to draw this conclusion. These conclusions would be greatly strengthened by presenting quantification data of the IP bands to demonstrate this decreased interaction quantitatively (as carried out in Figure 3I).

Minor concerns:

1. There are numerous grammatical errors. This manuscript would benefit from professional proof reading before resubmission.

2. The authors state that C42S mutation results in ROT4 diffusion into the cytoplasm. However, the data presented in Figure 1B and 1C do not agree with this statement. Biochemical (Figure 1C) and microscopical (figure 1B) evidence suggest that two ROT4 C42S pools now exist, a PM-bound fraction and a cytosolic fraction. Please can the authors amend the text.

3. Figure S2 shows ROT4 expression levels of the transgenic lines used in this study by qPCR. It is now generally considered best practice to show transgene expression by qRT-PCR. Therefore, please can the authors carry out qRT-PCR to quantitatively show ROT4 expression levels in the wild type compared to overexpression lines.

RESPONSE TO THE REVIEWERS

Referee #1:

Li et al. show that plasma membrane targeting of ROT4 is mediated by S-acylation at Cys42. They further show that overexpression of ROT4, but not of a ROT4 C42S mutant, causes massive upregulation of defense-related genes leading to enhanced resistance against bacterial (*P. syringae* DC3000) and fungal (*B. cinerea*) pathogens. Co-IP mass spec analysis identified the cytoplasmic receptor-like kinase BSK5 as an interactor of ROT4, and the interaction of ROT4 with BSK5 was found to be reduced in the C42S mutant. Further Co-IP experiments suggested that ROT4 attenuates the interaction between BSK5 and the pattern recognition receptor PEPR1. Based on these findings Li et al. propose a model of the mechanism underlying ROT4-dependent induction of immunity. Previous studies showed that the BSK5/PEPR1 complex dissociates upon ligand binding, resulting in the release of BSK5 and pattern-triggered immunity. According to the proposed model, ROT4 facilitates BSK5 release thereby promoting immune signaling.

The paper reveals a novel function of ROT4 in immune signaling, which relies on post-translational modification by S-acylation for targeting to the plasma membrane. ROT4 is proposed to facilitate the release of BSK5 from the pre-existing BSK5/PEPR1 complex, thereby activating pattern-triggered immunity. This is certainly a very interesting hypothesis. The topic is of broad interest and well within the scope of EMBO Reports. The data related to ROT4 localization, ROT4-dependent induction of defense, and S-acetylation at Cys42 as a prerequisite for both, are solid and convincing. However, further evidence is needed to substantiate the model proposed for the ROT4 mode of action.

RESPONSE: Thank you very much for your nice comments! Following your suggestions, we performed additional experiments and the point-to-point responses are shown below.

Major:

1. The authors state that ROT4-mediated 'dissociation of PEPR1 and BSK5 ... mimics the process in the presence of Pep1, activating downstream immunity pathways' (line 201-203). In my view, this is only partially true. In presence of Pep1, BSK5 is not only released but also phosphorylated by PEPR1 in the kinase activation loop. Phosphorylation and kinase activity were shown to be critical for the function of BSK5 in pattern-triggered immunity (Majhi et al. 2019). Therefore, the model implies that BSK5 is phosphorylated in the presence of an excess of ROT4. If this is the case, evidence for BSK5 phosphorylation in ROT4 overexpressing lines should be included. On the other hand, if phosphorylation of BSK5 is not required, the ROT4 overexpression phenotype would be expected in the background of a *bsk5* mutant complemented with a phospho-dead BSK5 variant.

RESPONSE: Thank you very much! Following your suggestion, we tried to detect the phosphorylation status of BSK5 in the ROT4 overexpressing lines. However, we found that phosphorylation of BSK5 was even undetectable in the WT plants. Indeed, the in vivo data of BSK5 phosphorylation in plant cells is also not shown in the previous study (Majhi et al. 2019), possibly resulted from the instability of phosphorylation on BSK5. Therefore, we used an alternative strategy to test the relationship between BSK5 phosphorylation and ROT4. The effect of ROT4 overexpression on the association between PEPR1 and the phosphorylation defective BSK5(S209A/T210A) in a co-IP assay. As a result, overexpression of ROT4 enhances the dissociation of BSK5(WT) and PEPR1, but does not affect the interaction between BSK5(S209A/T210A) and PEPR1 (Figure EV4A-EV4B). This data supported the notion that phosphorylation of BSK5 is essential for the ROT4-mediated BSK5-PEPR1 dissociation.

Consistently, Pep1 enhances the dissociation of BSK5(WT) and PEPR1, but does not affect the association between BSK5(S209A/T210A) and PEPR1 (Figure EV4C-EV4D), suggesting that the important of BSK5 phosphorylation for its dissociation from PEPR1. Collectively, these data showed an important relationship among ROT4, Pep1-PEPR1, and BSK5 phosphorylation. The description and discussion are also included in the revised manuscript.

2. Most of the results were obtained by overexpressing ROT4, or the ROT4 C42S mutant. I am concerned that the presence of excessively large amounts of ROT4 in the plasma membrane may lead to interactions that do not occur in wild-type plants. Therefore, strong support for the conclusions drawn could be obtained by including a loss-of-function experiment.

RESPONSE: Thank you very much! As we know, ROT4 belongs to the ROT-FOUR LIKE/DEVIL (RTFL/DVL) family with many conserved members in Arabidopsis, because of functional redundancy, overexpression is used in all previous works on this family of peptides. To answer your nice question, we included the phenotype data of plants with *ROT4* overexpression in the *PEPR1* mutant background. The result showed that overexpression of *ROT4* suppressed plant development in the WT background but not in the *PEPR1* mutant background (Figure EV5). Therefore, the excessive ROT4 is not enough to suppress plant development when the PEPR1 signaling is blocked, supporting the functional specificity of this peptide on the plasma membrane. Following your suggestion, we also included the phenotype data of the overexpression lines of two additional RTFL members. As a result, overexpression of an S-acylated member *RTFL18* suppressed plant growth, but overexpression of a non-S-acylated member *RTFL13* did not affect plant development (Figure EV2E, EV2F), supporting the function of S-acylation of RTFL members.

Minor:

1. Additional support for the model could be provided by comparing the ROT4-dependent and the BSK5-dependent transcriptomes - large overlap expected.

RESPONSE: Thank you very much! We showed the interaction between BSK5 and ROT4, but the previous study indicated that the *BSK5* mutant does not display a development defective phenotype, thus their functions may be predominantly overlapped in plant immunity. In the previous work, an immunity response gene *PRI* was showed to be upregulated in the *BSK5* mutant (Majhi et al. 2019). Similarly, our quantitative RT-PCR data showed that the expression of *PRI* was significantly increased in the *ROT4* overexpression plants (Figure 2D), consistent with their functional association in plant immunity. Following your suggestion, the similar expression pattern of *PRI* in the *BSK5* mutant and the *ROT4* overexpressing plants is also described in the revised manuscript. In the future study, the effect of ROT4 and BSK5 on global gene expression may be analyzed by further RNA-Seq analysis.

2. The title is somewhat enigmatic. I would prefer a title which names the main players/findings

RESPONSE: Thank you very much! Because we think the important point of our work is establishing the functional association between the non-secreted peptide ROT4 and the secreted plant elicitor peptide signaling, the current title may be more interesting to general readers. Certainly, we can change the title if you think it is necessary.

3. The heat map in Figure 2C is said to confirm upregulation of genes. The heat map is just another way of presenting the data, rather than a confirmation of the results

RESPONSE: Thank you! Following your suggestion, the description of Figure 2C has been clarified in the revised manuscript.

4. The conclusion that 713 proteins '...were specifically associated with GFP-ROT4' (line 168) is not justified. The list of 713 proteins is likely to contain many that do not interact with GFP-ROT4. Many more biological replicates and statistics would be required to justify this conclusion. Aiming at the identification of proteins interacting at the plasma membrane, GFP-ROT4-C42S would have made a better control than GFP alone.

RESPONSE: Thank you very much! The candidate proteins are from IP-mass spectra data for potential ROT4-interacting proteins in two biological replications. The detailed information and analysis of these candidate proteins are included in the Appendix Table S3. The mass spectra data provided a clue for our study on the functional association between BSK5 and ROT4. Indeed, the detailed data of mass spectra is not necessary for our main conclusions, but will provide an important resource for further research on ROT4. Therefore, we moved this proteomics analysis to Appendix Figure S3.

5. Please describe the principle of the biotin-switch assay, either in the text or in the methods section. A few sentences will suffice (similar how you did it in your previous publication, Liu et al. 2023)

RESPONSE: Thank you! Following your suggestion, the principle of the biotin-switch assay is included in the revised manuscript.

6. I do not understand the experiment shown in Figure 4C,D. Why were the peptides added? No peptides were needed in Figure 4B? Why do you get the same effect for flg22 as for Pep1?

RESPONSE: Thank you very much! Because immunity responses is constitutively activated in the *ROT4(WT)* overexpressing lines, we think Pep1 is not necessary for its disease resistance. Therefore, in the Pst DC3000 infection assay (Figure 4B), we did not include Pep1 and the data supported the function of ROT4 S-acylation in bacterial resistance. In the *Botrytis cinerea* infection assay, flg22 was usually used for immunity induction, thus we used it in the control samples. The data showed that Pep1 and flg22 treatment showed similar patterns, supporting that overexpression of *ROT4* improves disease resistance. The description has been clarified in the revised manuscript.

7. Hurst et al 2023, Current Biology 33:1588 should be discussed

RESPONSE: Thank you! The work (Hurst et al 2023) has been cited in the revised manuscript.

8. There are many unusual abbreviations: TCEP, MMTS, HPDP, ... please spell out.

RESPONSE: Thank you! The abbreviations have been spelled out.

9. Fig. 3H is not really convincing. Yes, the band is weaker when both PEPR1-GFP and Pep1 are present, but this is also observed in the input panel.

RESPONSE: Thank you very much! Following your suggestion, we replicated the experiments. The new immunoblot images and quantification data from three biologically independent experiments are shown in the revised manuscript.

Referee #2:

In this manuscript entitled *S*-acylation of a non-secreted peptide controls plant immunity via secreted-peptide signal activation by Wenliang Li, Tushu Ye and colleagues the authors address the importance of acylation as a posttranslational modification of the non-secreted ROTUNDIFOLIA4 (ROT4) peptide in Arabidopsis. The ROT4 peptide belongs to the RTFL/DEVIL (DVL) peptide family that consist of 23 members in Arabidopsis and that has a conserved domain of about 30 amino acids. The authors of this work use CSS-PALM analysis to identify potential sites of *S*-acylation (cysteine residues) in the conserved domain of several ROT members, there among ROT4. They explore the functional importance of this *S*-acylation site in ROT4 by several methods and show that *S*-acylation is important for another member of the RTFL/DEVIL family (RTFL18). From their experiments the authors conclude that the *S*-acylation is important for membrane localization and for the interaction between ROT4 and the cytoplasmic kinase BSK5. Using overexpressing (OE) lines the authors conclude that ROT4 is important for immunity responses in Arabidopsis by attenuating the dissociation of the immune LRR-RLK receptor PEPR1 and BSK5, a disassociation that is also triggered when PEPR1 binds its peptide partner, the secreted Pep1 peptide. Furthermore, the authors show through pathogen assays that OE ROT4 lines are less susceptible to *Pseudomonas syringae* pv. tomato DC3000 and the fungal pathogen *Botrytis cinerea* in combination with the pathogenic peptide flg22 or Pep1.

The manuscript is clear and concise although it would benefit from some additional language editing. The figures are of good quality and the results are clearly presented. In most cases it is indicated how many independent times the experiments were repeated and proper statistical analysis are performed. Critical references are given, and the M and M section is adequately written.

RESPONSE: Thank you very much for your comments! Following your suggestion, the manuscript has been edited by a native English specialist.

This manuscript reports two key findings:

- The importance of acylation for the function of at least two RTFL family members and the proposed function of ROT4 in plant immunity.

This manuscript adds novelty to the importance of posttranslational modification of peptides although it is already known that acylation of receptors in plants stabilizes ligand-induced receptor kinase complex formation during immune signaling (<https://www.sciencedirect.com/science/article/pii/S0960982223002385>). The work in this manuscript will therefore be of interest to the plant molecular biology community.

The finding that acylation is important for the membrane localization of ROT4 and for its OE developmental phenotypes is well founded. However, the conclusion drawn on the role of ROT4 in plant immunity requires further experimental work to be conclusive.

RESPONSE: Thank you very much for your nice comments! Following your suggestions, we performed additional experiments and the point-to-point responses are shown below.

Below follows a list of comments and concerns and some suggestions that could improve the manuscript.

Major concerns:

- All the transgenic experiments performed for the OE lines are done in the Col-0 WT background in the presence of the endogenous ROT4 gene, so in the lines where conclusions are drawn for the OE of ROT4 with a cysteine to serine substitution the endogenous ROT4 is still active. In my opinion the results would be more convincing if this was done in a rot4 mutant background. For example when comparing the RNA seq data and QPCR data between the two versions of ROT4 OE for the C to S

substitution the active ROT4 peptide is still present.

RESPONSE: Thank you very much! As we know, ROT4 belongs to the ROT-FOUR LIKE/DEVIL (RTFL/DVL) family with many conserved members in Arabidopsis, because of functional redundancy, overexpression is used in all previous works on this family of peptides. Following your suggestion, we performed quantitative RT-PCR and found that the transcript level of *ROT4* was much higher in the overexpressing lines than in the WT control plants (Appendix Figure S1). Therefore, we think that the endogenous ROT4 would only have a very small effect. To confirm the functional specificity of *ROT4* overexpression, we also included the phenotype data of plants with *ROT4* overexpression in the *PEPRI* mutant background. The result showed that overexpression of *ROT4* suppressed plant development in the WT background but not in the *PEPRI* mutant background (Figure EV5), supporting the specific effect of *ROT4* overexpression on plant development.

- For the experiment where RTFL13 was tagged with GFP to show "global" expression in the cell and its lack of S-acylation a more convincing experiment for the essential role of this cysteine for S-acylation would be to include results where the Arginine is substituted for a Cysteine to see if this would restore plasma membrane localization and S-acylation. Also, it would be interesting to compare OE phenotypes between RTFL13 and the other members of this family and see if the change of Arginine to Cysteine would affect the phenotypes.

RESPONSE: Thank you! Following your nice suggestion, we mutated the arginine residue to cysteine residue and analyzed its effect on the localization of RTFL13. The confocal and cell fractionation data indicated that the mutation increases the member localization of RTFL13 (Figure EV2B, EV2C), supporting the contribution of the conserved cysteine residue in the regulation of membrane association. Following your advice, we also included the phenotype data of the

overexpression lines of two additional RTFL members. The results indicated that overexpression of an S-acylated member *RTFL18* suppressed plant growth, but overexpression of a non-S-acylated member *RTFL13* did not affect plant development (Figure EV2E, EV2F), supporting the function of S-acylation on RTFL members.

- From the results from the RNA seq experiments it is not clear to me why the authors choose to focus their follow up experiments in immunity and BSK5. From the GO analysis it would appear that OE ROT4 also induces responses compatible with biotic stresses. A better line of reasoning needs to be made. The authors conclude that the reduced growth rate of the OE ROT4 lines are compatible with an elevated and continuous immune response, but there are many other upregulated genes that could affect the growth phenotype. Also there is no inhibition in root length in the ROT4 OE lines unless Pep1 is added while the model proposes that ROT4 on its own, when OE, activates immunity. In general I think the authors need to be more cautious with drawing conclusions on the role of ROT4 regulating plant immunity based on OE experiments. I follow the authors concern with regard to redundancy among the RTFL/DEVIL family members but CRISPR lines where ROT4 (and other members with overlapping expression patterns) are mutated should be investigated for susceptibility to pathogens to draw more convincing conclusions.

RESPONSE: Thank you very much! Based on the predominant enrichment of upregulated genes in immunity and biotic responses in the RNA-seq data (Figure 2B), we think that the growth phenotype may be dependent on constitutive activation of immunity responses. We further showed that overexpression of *ROT4* suppressed plant development in the WT background but not in the *PEPRI* mutant background (Figure EV5), supporting that the effect of ROT4 overexpression on development is dependent on the plant elicitor peptide immunity signaling. Following your suggestion, the description and conclusion are revised with more cautions.

- With regards to the RNAseq experiment it is not clear to me why the ROT4 OE substitution line was not included and only used to later check for selected candidate genes by QPCR. This needs to be addressed.

RESPONSE: Thank you very much! Because when we initiated our research, we only would like to know the pathways regulated by ROT4, thus the C42S mutant lines did not be included for RNA-seq. When obtaining the RNA-seq data, we found many immunity genes are upregulated in the *ROT4(WT)* overexpressing plants. Therefore, we focused on the expression of these genes and used individual lines of the *ROT4(WT)* and *ROT4(C42S)* overexpressing plants for further quantitative RT-PCR analysis. Indeed, the transcript levels of most candidate genes are similar in the wild-type, vector control, and *ROT4(C42S)* overexpressing plants (Figure 2D). The description has been clarified in the revised manuscript.

- In line 175 the authors conclude that BSK5 interacts with the membrane localized ROT4, however, apart from the split YFP experiments there is no in vivo evidence for this interaction. Given that ROT4 localizes nicely to the PM in root tissue it should be possible to do FRET or even better FRET-FLIM (for quantification given that the IP data only shows a reduction in interaction with the ROT4 substituted lines) to show in vivo interactions between BSK5 and ROT4 (and reduced interaction in the substituted lines). Ideally since the authors suggest that Pep1 mediates this interaction after binding PEPR1 the in vivo experiments should be done with and without the presence of exogenous Pep1 (again this should be possible to do in root tissue).

RESPONSE: Thank you! Besides the BiFC assay, we used in vivo co-IP to confirm the BSK5-ROT4 interaction and the effect of ROT4 S-acylation on this association (Figure 3A-3C). Following your suggestion, we additionally performed a FRET assay to detect BSK5-ROT4 association in plant cells. The

quantitative FRET data confirmed the interaction between BSK5 and ROT4, as well as S-acylation of ROT4 mediates its efficient association with BSK5 (Figure 3D). The BifC data are moved to Figure EV3. Following your nice advice, we also detect the effect of Pep1 on the association between BSK5 and ROT4. The data showed that Pep1 enhances the BSK5-ROT4 interaction in plant cells (Figure 3E, 3H), consistent with our model.

- Line 241: The final conclusion of this work where the authors conclude that ROT4 connects secreted and non-secreted signaling in plant immunity is not shown. The authors show that there is less interaction between BSK5 and PEPR1 in the ROT4 OE lines and that there is less interaction between BSK5 and ROT 4 OE (C42S) lines but there is no experiment where the interaction between BSK5 and ROT4 is measured in response to Pep1.

RESPONSE: Thank you very much! Our data showed that Pep1 induces the dissociation of PEPR1-BSK5, whilst ROT4 interacts with BSK5 and also enhances the dissociation of PEPR1-BSK5, suggesting the connection between ROT4 and Pep1-PEPR1 signaling. Following your suggestion, we detected the effect of Pep1 on the BSK5-ROT4 interaction and the data showed that Pep1 enhances the BSK5-ROT4 interaction in plant cells (Figure 3E, 3H). Therefore, Pep1 reduces the BSK5-PEPR1 interaction but increase the BSK5-ROT4 interaction. As a model, Pep1 binds to PEPR1 and releases BSK5 for the efficient BSK5-ROT4 interaction; overexpression of ROT4 enhances the dissociation of PEPR1-BSK5 for constitutive activation of immunity.

Minor comments:

Line 46: Previous studies showed that activation expression of the member of this family in Arabidopsis, such as ROTUNDIFOLIA4 (ROT4) or DEVIL1 (DVL1)/RTFL18, results in a phenotype with small and round leaves (Narita et al,

2004; Wen et al, 2004). Please explain what you mean by "activation expression"

RESPONSE: Thank you! In the first paper on ROT4, the authors isolated a mutant with upregulation expression of ROT4 in T-DNA activation lines and the phenotypes were then confirmed by ROT4 overexpression. Following your suggestion, we think “overexpression” may be easier to be understood and it has been revised in the manuscript.

Line 62: Although PLANT ELICITOR PEPTIDES (PEPs) were expressed without an N terminus signaling motif, the mature forms of these peptides are secreted outside and recognized by PEP RECEPTOR1/2 (Bartels & Boller, 2015). What is the reasoning for stating that the PEPs were expressed without an N terminal signaling peptide?

RESPONSE: Thank you! Most secreted peptides are transported outside plant cells via conventional protein secretion mediated by their N terminus signaling peptide motifs, while PEPs do not contain an N terminus signaling motif. Previous studies showed that PEPs are secreted to extracellular regions, but the secretion mechanism is unknown. Following your suggestion, the description is clarified in the manuscript.

Line 96: please make an additional supplementary figure where you show the whole ROT4 protein with all the potential cysteine sites that are identified as potential S-acylation sites.

RESPONSE: Thank you! Following your suggestion, the image of a whole ROT4 protein with all potential cysteine sites is included Figure EV1A.

Line 109: The further cell fractionation data indicated that most of the wild-type

GFP-ROT4 accumulated.... Please rewrite.

RESPONSE: Thank you! The sentence has been revised.

Line 116: Because the Arabidopsis RTFL family contains 23 members, it is interesting to analyze the C42 site of ROT4 is conserved among the RTFL proteins.... Unclear, please rewrite.

RESPONSE: Thank you! The sentence has been revised.

Line 137: unknown. To catch the hints on the pathways regulated by ROT4... please rewrite

RESPONSE: Thank you! It has been revised.

Line 139: In several figures, there among figure S2 the authors use the term vector. Please specify what is meant with this. I presume that these are lines expressing the vector used to make the OE lines with the GFP tag.

RESPONSE: Thank you! Yes, they are free *GFP* overexpressing lines using the vector for transformation. The description on vector controls has been revised in the figure legends.

Figure 2C pathogen is misspelled. Also there is some confusion to using the line numbering here instead of indicating that it represents 2 independent experiments since the S2 only shows 2 lines. It is also confusing to refer to figure S2 with regards to the RNAseq experiments since the OE ROT4 substitution lines were not used for RNA seq only to compare QPCR data.

RESPONSE: Thank you! The word “pathogen” in Figure 2C has been corrected. The numbers here indicated 3 independently repeated experiments and it has been revised in Figure 2C. We mentioned on Figure S2 (Revised Appendix Figure S1) to show the overexpression of *ROT4(WT)*. Following your suggestion, the description has been clarified.

Line 165: Here the authors use the term GFP plants, other places vector. Please be concise in the terminology used to avoid confusion.

RESPONSE: Thank you! It has been revised.

Referee #3:

The key finding of this work is that the conserved S-acylation/S-palmitoylation site Cys42 in ROT4, a non-secreted peptide, is required for activation of plant immune responses through interaction with PEPR1, a receptor-like kinase in Arabidopsis. The role of S-acylation in non-secreted peptides has not yet been investigated and this work will therefore be of interest to both the fields of S-acylation and plant immunology.

The data presented in Figure 4 are excellent and highly compelling - These data demonstrate at a whole organism level how loss of the ROT4 Cys42 S-acylation state impacts plant growth, pathogen resistance and root length. However, the biochemical approaches taken in this work to understand this phenotype, as demonstrated in Figures 1-3, will require strengthening prior to publication.

While this research is novel in its findings, linking the activity of secreted and non-secreted peptides by S-acylation to activate plant immunity, there are several concerns that I would like the authors to address prior to publication.

RESPONSE: Thank you very much for your nice comments! Following your suggestions, we performed additional experiments and the point-to-point responses are shown below.

Major concerns:

1. Based on the data in Figure 1A, the authors state that C42 is the predominant ROT4 S-acylation site. The authors do not clarify whether other cysteines are contributing to this S-acylation state. It may be the case that incomplete MMTS blocking, resulting in the bands appearing in the NH₂OH (-) lane, is masking that C42 is the only S-acylation site. Please could the authors present western blots where there are no bands in the NH₂OH (-) lanes. If this is not possible, please can the authors present quantified data of the reduction in S-acylation. This would clarify whether C42 is the only S-acylation site or is the predominant contributor to ROT4 S-acylation state.

If C42 is not the only S-acylation site, the authors may have to consider repeating the experiments presented here with a double cysteine mutant to show how total loss of S-acylation impacts ROT4 function, which I appreciate will be a lot of work.

RESPONSE: Thank you very much! Following your nice suggestion, we performed a quantification of S-acylation for the WT and cysteine mutant ROT4 proteins from three biologically independent experiments. As the data shown in Figure 1B, compared to ROT4(WT), the S-acylation level of ROT4(C42S) is almost lost; no significance is shown between the S-acylation levels of WT and other cysteine variants. These data supported that C42 is the predominant S-acylation site of ROT4.

2. Figure 1C shows ROT4 and ROT4 C42S membrane fractionation by ultracentrifugation, however, no loading controls are presented. PM and cytoplasmic protein contamination markers should also be included. Please could the authors

include loading controls and contamination markers for this experiment.

RESPONSE: Thank you very much! Indeed, we always used free GFP as a marker of cytoplasm fraction and PAT12-GFP (a protein S-acyltransferase on the plasma membrane) as a marker of membrane fraction in each cell fractionation experiment and we did not show the controls in the previous submission. Following your suggestion, the cell fractionation data of GFP and PAT12-GFP are included in Figure EV2A.

3. The authors state that the ROT4 C42S mutation reduces ROT4-BSK1 interaction. However, the IP data presented in Fig 3D and 3E are currently not sufficiently compelling to draw this conclusion. These conclusions would be greatly strengthened by presenting quantification data of the IP bands to demonstrate this decreased interaction quantitatively (as carried out in Figure 3I).

RESPONSE: Thank you very much! Following your suggestion, the IP results in Figure 3A and 3B (previous Figure 3D and 3E) were quantified using the data from three biologically independent experiments. The quantitative data are shown in Figure 3C. In addition, we also confirmed the effect of S-acylation on the ROT4-BSK5 interaction in a FRET assay (Figure 3D). These quantified data supported the notion that S-acylation of ROT4 is critical for its interaction with BSK5.

Minor concerns:

1. There are numerous grammatical errors. This manuscript would benefit from professional proof reading before resubmission.

RESPONSE: Thank you! Following your suggestion, the manuscript has been edited by a native English specialist.

2. The authors state that C42S mutation results in ROT4 diffusion into the cytoplasm. However, the data presented in Figure 1B and 1C do not agree with this statement. Biochemical (Figure 1C) and microscopical (figure 1B) evidence suggest that two ROT4 C42S pools now exist, a PM-bound fraction and a cytosolic fraction. Please can the authors amend the text.

RESPONSE: Thank you! Following your suggestion, the description has been revised.

3. Figure S2 shows ROT4 expression levels of the transgenic lines used in this study by qPCR. It is now generally considered best practice to show transgene expression by qRT-PCR. Therefore, please can the authors carry out qRT-PCR to quantitatively show ROT4 expression levels in the wild type compared to overexpression lines.

RESPONSE: Thank you very much! Following your suggestion, we used quantitative RT-PCR to detection the levels of *ROT4* in the control and overexpression lines. The data are included in Appendix Figure S1.

Dear Prof. Lai,

Thank you for the submission of your revised manuscript to our editorial offices. I have now received the reports from the three referees that I asked to re-evaluate your study, you will find below. As you will see, the referees now fully support the publication of the study in EMBO reports.

Before I can proceed with formal acceptance, I have these editorial requests I ask you to address in a final revised manuscript:

- Please provide the abstract written in present tense throughout.
- Please add scale bars of similar style and thickness to all the microscopic images (main, EV and Appendix figures), using clearly visible black or white bars (depending on the background). Please place these in the lower right corner of the images themselves. Please do not write on or near the bars in the image but define the size in the respective figure legend. Presently many scale bars are rather small/thin and hard to see.
- Please make sure that the number "n" for how many independent experiments were performed, their nature (biological versus technical replicates), the bars and error bars (e.g. SEM, SD) and the test used to calculate p-values is indicated in the respective figure legends (for main, EV and Appendix figures) of the final revised manuscript. Please also check that all the p-values are explained in the legend, and that these fit to those shown in the figure. Please provide statistical testing where applicable. Please avoid the phrase 'independent experiment', but clearly state if these were biological or technical replicates. Please also indicate (e.g. with n.s.) if testing was performed, but the differences are not significant. In case n=2, please show the data as separate datapoints without error bars and statistics. See also:
<http://www.embopress.org/page/journal/14693178/authorguide#statisticalanalysis>

- Please format the figure legends according to our journal style. See the respective section in our guide to authors (please find the link below). Please separate each panel description by a line brake and make sure that the panels are listed in alphabetic order. Moreover, please add to each legend a 'Data Information' section explaining the statistics used or providing information regarding replicates and scales.

- We now use CRediT to specify the contributions of each author in the journal submission system. CRediT replaces the author contribution section. Please use the free text box to provide more detailed descriptions and do not provide your final manuscript text file with an author contributions section. See also our guide to authors:
<https://www.embopress.org/page/journal/14693178/authorguide#authorshippinguidelines>

- Please remove the section 'Supporting data' (listing the EV and Appendix items) from the manuscript text file.
- There is one excel file with 5 Appendix tables uploaded. Tables S1, S2, S3 and S4 are datasets. Please upload these as single/separate excel files named Dataset EV1, Dataset EV2, Dataset EV3 and Dataset EV4. Please add a title and a legend on the first TAB of each of these files. Finally, please change the callouts of these items in the manuscript text file (Dataset EVx).
- Please add Table S5 (primers used) to the Appendix file. Please name this Appendix Table S1, add a legend to the table in the Appendix and include it into the table of contents. Finally, please change the callouts for this table accordingly.
- Please remove all the author information and affiliations from the Appendix title page. Just state there 'Appendix for: S-acylation of a non-secreted peptide controls plant immunity via secreted-peptide signal activation' followed the table of contents (TOC) with page numbers. Please also move the legends of the Appendix items below each item (not on the next page). This will render the Appendix more comprehensible.

In addition, I would need from you:

- a short, two-sentence summary of the manuscript (not more than 35 words).
- two to four short (!) bullet points highlighting the key findings of your study (two lines each).
- a schematic summary figure that provides a sketch of the major findings (not a data image) in jpeg or tiff format (with the exact width of 550 pixels and a height of not more than 400 pixels) that can be used as a visual synopsis on our website.

Best,

Referee #1:

The authors have made every effort to revise their manuscript along the lines suggested by me and by the other anonymous reviewers. Whenever one of the proposed experiments was not feasible, they came up with alternative ways to address the critical points. Quite some additional data have been included, and they all support the conclusions that had been drawn, and the model presented in the original manuscript. I am fully satisfied by the authors' responses to the points I had raised, and I am quite impressed by the revisions they made, which resulted in a much improved manuscript.

Referee #2:

I have reviewed a previous version of this manuscript and I am content with the present revised version. The authors have addressed all my concerns and comments in a thorough way and I am grateful for the effort they placed in the revision.

Referee #3:

This paper shows that that the conserved site of S-acylation (Cys42) in ROT4, a non-secreted peptide, is the predominant site of ROT4 S-acylation. Cys42 is required for ROT4 plasma membrane localisation and activation of plant immune responses through its interaction with PEPR1, a receptor-like kinase, in Arabidopsis.

The authors present data showing that overexpression of ROT4 wt but not the ROT4 S-acylation mutant leads to increased resistance to *P.syringae* and *B.cinerea* pathogens by activating PTI signalling through enhancing the dissociation of BSK5 from the BSK5/PEPR1 complex.

In my opinion, all of the concerns raised have been addressed. Thank you to the authors for addressing all of the reviewers' comments.

All editorial and formatting issues were resolved by the authors.

Prof. Jianbin Lai
South China Normal University
School of Life Science
55 Zhongshan Avenue West, Tianhe
Guangzhou, Guangdong 510631
China

Dear Prof. Lai,

I am very pleased to accept your manuscript for publication in the next available issue of EMBO reports. Thank you for your contribution to our journal.

Yours sincerely,
